# Dose estimates and their uncertainties for use in epidemiological studies of radiation-exposed populations in the Russian Southern Urals

**Elena A. Shishkina[1,2], Bruce A. Napier[3] \*, Dale L. Preston[4], Marina O. Degteva[1]†**

**1** Biophysics Laboratory, Urals Research Center for Radiation Medicine, Chelyabinsk, Russia,
**2** Chelyabinsk State University, Chelyabinsk, Russia, **3** Energy and Environment Directorate, Pacific Northwest National Laboratory, Richland, Washington, United States of America, **4** Hirosoft International LLC, Eureka, California, United States of America

☯ These authors contributed equally to this work.
† Deceased.
\* BruceAlanNapier@gmail.com

**Data Availability Statement:** Russian privacy laws prohibit the release of potentially identifiable individual data and therefore the personal information, which is important for dose and dose

## Abstract

Many residents of the Russian Southern Urals were exposed to radioactive environmental pollution created by the operations of the Mayak Production Association in the mid- 20$^{th}$ century. There were two major releases: the discharge of about $1 \times 10^{17}$ Bq of liquid waste into the Techa River between 1949 and 1959; and the atmospheric release of $7.4 * 10^{16}$ Bq as a result an explosion in the radioactive waste-storage facility in 1957. The releases into the Techa River resulted in the exposure of more than 30,000 people who lived in riverside villages between 1950 and 1961. The 1957 accident contaminated a larger area with the highest exposure levels in an area that is called the East Urals Radioactive Trace (EURT). Current epidemiologic studies of the exposed populations are based on dose estimates obtained using a Monte-Carlo dosimetry system (TRDS-2016MC) that provides multiple realizations of the annual doses for each cohort member. These dose realizations provide a central estimate of the individual dose and information on the uncertainty of these dose estimates. In addition, the correlation of individual annual doses over realizations provides important information on shared uncertainties that can be used to assess the impact of shared dose uncertainties on risk estimate uncertainty.This paper considers dose uncertainties in the TRDS-2016MC. Individual doses from external and internal radiation sources were reconstructed for 48,036 people based on environmental contamination patterns, residential histories, individual $^{90}$Sr body-burden measurements and dietary intakes. Dietary intake of $^{90}$Sr resulted in doses accumulated in active bone marrow (or simply, marrow) that were an order of magnitude greater than those in soft tissues. About 84% of the marrow dose and 50% of the stomach dose was associated with internal exposures. The lognormal distribution is well-fitted to the individual dose realizations, which, therefore, could be expressed and easily operated in terms of geometric mean (GM) and geometric standard deviation (GSD). Cohort average GM for marrow and stomach cumulative doses are 0.21 and 0.03 Gy, respectively. Cohort average dose uncertainties in terms of GSD are as

uncertainty calculation (such as residence histories, birthdate, sex) cannot be made available. The large number of MC simulations (~1 terabyte datasets) were significant for uncertainty estimation but are hardly of interest in themselves. In the paper we tried to present the statistics in detail. If readers are interested in more detailed summary statistics, a non-author contact for these inquiries is the URCRM Ethics Officer, Sergey Shalaginov - shalaginov@urcrm.ru.

**Funding:** EAS: Federal Medical-Biological Agency of Russia Contract N 27.501.19.2 in the framework of Russian Federal Targeted Program "Provision of nuclear and radiation safety for the period 2016-2020 and for the period up to 2035". https://xn——btb4bfrm9d.xn–p1ai/ and US Department of Energy, Project Joint Coordinating Committee for Radiation Effect Research (JCCRER) dose reconstruction for the Urals. https://www.energy.gov/ehss/international-health-studies-and-activities BAN: PNNL Contract DE-AC05-76RL01830, (US Department of Energy), Project JCCRER DOSE RECONSTRUCTION FOR THE URALS, Budget and Reporting Number HS0240030, https://www.energy.gov/ehss/international-health-studies-and-activities DLP: University of Southern California Prime Award # DE-HS0000091 (US Department of Energy), Project Epidemiological and Biostatistical Assistance for Project 2.2 Mayak Worker Cancer Mortality and for Project 1.2 Techa River Cohort Cancer mortality and Incidence, USC Subaward 122032572 – Mod 3, https://keck.usc.edu and https://www.energy.gov/ehss/international-health-studies-and-activities MOD (deceased): Federal Medical-Biological Agency of Russia Contract N 27.501.19.2 in the framework of Russian Federal Targeted Program "Provision of nuclear and radiation safety for the period 2016-2020 and for the period up to 2035". https://xn——btb4bfrm9d.xn–p1ai/ and US Department of Energy, Project Joint Coordinating Committee for Radiation Effect Research (JCCRER) dose reconstruction for the Urals. https://www.energy.gov/ehss/international-health-studies-and-activities The funders had no role in study design, data collection and analysis, decision to publish, or preparation of the manuscript.

**Competing interests:** The authors have declared that no competing interests exist.

follows: for marrow it is 2.93 (90%CI: 2.02–4.34); for stomach and the other non-calcified tissues it is 2.32 (90% CI: 1.78–2.9).

## Introduction

Epidemiological studies of radiation exposure effects need dosimetric support. The requirements for the dosimetry, according to [1], are as follows: 1) calculation of absorbed doses in organs (instead of effective doses); 2) description of the dose dynamics; 3) individual rather than group-averaged doses are preferred; and 4) estimates should be as accurate as possible. In uncontrolled radiation situations, as in the case of radioactive contamination of the environment, strict compliance with all these requirements is impossible. Doses are reconstructed by simulating the transport of radionuclides in the environment considering population-average behavioral and dietary habits, general patterns of radionuclide biokinetics and physical processes of radiation transport. Uncertainties in individual dose estimates can be described as shared and unshared uncertainties. Unshared uncertainties can arise, for example, because an individual's behavioral and dietary habits differ from the population average values assumed in the dosimetry system. Shared uncertainties arise because, for example, the population average values are estimates that differ from the actual values. Other sources of shared uncertainties include uncertainties in biokinetic model parameters, and imprecision in the characterization of the composition and timing of the releases. Shared uncertainties account for a large fraction of the uncertainty in individual dose estimates. Traditionally, risk estimation has considered only uncertainties associated with statistical variability in the outcome data, however in recent decades it has been recognized that characterization of the uncertainty in risk estimates should also consider the impact of dose uncertainty [2]. Dose uncertainty (particularly shared uncertainty) can result in biased risk estimates and underestimation of the risk estimate uncertainty. Proper characterization of the uncertainty in risk estimates is important in assessing the statistical significance of the risk estimates and understanding the range of risks that are consistent with the data, which can be important in risk management. Confidence intervals are also important when comparing results of studies from different populations. To date, only a few studies of radiation effects have considered the quantitative impact of dose uncertainty, particularly shared uncertainties, on risk estimates [3–6].

Modern dosimetric systems should provide both central estimates of individual doses and information on the uncertainty in these estimates. In particular, uncertainty information is needed for the epidemiological studies of the populations of the Russian Southern Urals exposed to radiation from radioactive environmental contamination that arose from the mid-20th century discharges of radioactive material from the Mayak Production Association (MPA). Population exposures were a consequence of two major releases: the discharge of about $1 \times 10^{17}$ Bq of liquid waste into the Techa River [7–9] between 1949 and 1956 and the atmospheric release of $7.4 * 10^{16}$ Bq following an explosion in a radioactive waste-storage facility in 1957 (sometimes called the "Kyshtym Accident") that formed the East Urals Radioactive Trace (EURT) [9–13]. Members of the exposed populations received both external and internal radiation exposures with bone-seeking long-lived $^{90}$Sr making a significant contribution to the internal dose.

Epidemiological studies in the Urals region provide a unique opportunity to quantify the long-term effects of chronic, low-dose-rate exposure in a large, unselected population based on follow-up of the Techa River Cohort (TRC) and the EURT Cohort (EURTC). Earlier risk

analyses have considered radiation effects in these two cohorts separately. However, since both cohorts consist of rural residents of the same region with similar socio-economic status, similar levels of medical care, and similar ethnic composition, it was recently decided to combine the two cohorts. Risk analyses in the combined cohort, which includes 47,950 of the 48,036 individuals with TRDS-2016 dose realizations, will have more statistical power than those carried out in the separate populations. About 5000 people in the combined cohort were exposed from both the Techa River contamination and the releases from the 1957 accident. It should be noted that, at this time, the combined cohort does not include people who were born after the initial releases, i.e. 1950 for Techa River residents and September 29, 1957 for those exposed as a consequence of the 1957 accident.

Epidemiological studies are based on a dosimetry system [14] that provides both a central estimate as well as uncertainty of individual doses. Radiation doses for Urals residents have been calculated with a set of computer codes known as the Techa River Dosimetry System (TRDS). The deterministic version (TRDS-2016D) [14] has been superseded by the stochastic (TRDS-2016MC) system, which uses the same basic equations and the same mean values of model parameters but provides 1,500 realizations of annual internal and external organ doses for each cohort member. The TRDS-2016MC uses a two-dimensional Monte Carlo method [15] accounting for both shared uncertainties and independent unshared uncertainties. This paper considers dose uncertainties calculated with TRDS-2016MC for the members of the combined TRC and EURTC cohorts.

The output of the Monte Carlo calculations is a set of individual annual external and internal doses separately calculated for the Techa River and EURT exposures over as much as 67 years to 23 organs for the 48,036 individuals. When describing the results of stochastic modeling, it should be noted that the sources and the dynamics of dose formation for organs other than the active marrow are similar, while the internal exposure to bone-seeking $^{90}$Sr is a significant, or even primary, source of marrow exposure. Therefore, the results presented here focus on doses to the stomach, as a representative soft tissue dose, and to the active marrow.

## Methods: TRDS-2016MC dosimetry system

In this section we outline the methods used to compute doses from internal and external exposures by members of the combined TRC and EURTC and describe the nature of the uncertainties in the dosimetric model parameters assumed for the Monte-Carlo dose computations.

The TRDS performs dose calculations by summing 4 dose components, viz: Techa River internal exposure; Techa River external exposure; EURT internal exposure, and EURT external exposure.

For simplicity in this description, we assume that individuals stay in one location throughout the follow-up period. However, the actual dose computations allow for people to live in multiple locations for a given year. More detailed descriptions of the dosimetric models are given in [14, 16].

### Techa River doses

The doses received by Techa River residents arose from internal exposure to eight radionuclides ($^{89}$Sr, $^{90}$Sr, $^{95}$Zr, $^{95}$Nb, $^{103}$Ru, $^{106}$Ru, $^{137}$Cs, and $^{144}$Ce) primarily from the ingestion of water and from external exposure to radiation from environmental contamination of riverside areas.

**Techa River internal dose.** The Techa River internal exposures arise from the ingestion of contaminated materials including water, milk, and, to a lesser extent, other foods. Estimates of annual intakes were based on time-dependent in-vivo measurements of $^{90}$Sr combined with

information on the composition and timing of the initial releases and environmental models of the transport of the radioactive material down the river. The intake of radioisotope $r$ in year $y$ for person $i$ born in year $b_i$ of age at time of intake $\tau_i(y) = y - b_i$ living in location $L_i(y)$ can be written as (Eq 1):

$$I_i^{T,r}(L_i(y), \tau_i(y)) = I_y^{90Sr} \times f_{L_i(y)}^{90Sr} \times \alpha^{90Sr}(\tau_i(y)) \times \xi_i \times R_{L_i(y)}^{r:90Sr} \qquad (1)$$

that depends on the following functions:

$I_y^{90Sr}$ is the $^{90}$Sr intake in year $y$ for a defined reference settlement;

$f_{L_i(y)}^{90Sr}$ is a ratio of the $^{90}$Sr intake at location $L_i(y)$ to that in the reference settlement;

$\alpha^{90Sr}(\tau_i(y))$ is the ratio of the $^{90}$Sr intake in children to that in adults;

$\xi_i$ is an individualized $^{90}$Sr intake scaling factor that depends on the availability of individual or household $^{90}$Sr measurements;

and

$R_{L_i(y)}^{r:90Sr}$ is a time- and location-dependent ratio used to convert $^{90}$Sr intake to intake of radioisotope $r$.

The average-intake functions for the reference villages, $I_y^{90Sr}$, (Eq 1), were the basis of the Techa River internal dose reconstruction [17]. The uncertainty of these functions was a *significant* contributor to the uncertainty of the Techa River internal doses [16]. These reference functions were developed for two villages–Metlino and Muslyumovo (7 km and 78 km from the site of releases, respectively). In Metlino, the use of the Techa River as a source of water supply was prohibited in mid-August 1951. In 1953 and 1954, wells were constructed in all other Techa riverside villages to minimize the consumption of river water. Thus, Metlino residents were supplied with non-contaminated water since August 1951, and in 1956 they were evacuated to non-contaminated areas, primarily New Metlino, a village created for them, the Techa River village that was closest to Ozyorsk and the Mayak facility. As a result, the period of $^{90}$Sr intake was relatively short in Metlino. The temporal pattern of $^{90}$Sr intake in Muslyumovo, which was never evacuated, is typical for most other Techa River villages prior to their evacuation date (if any). The time-dependent $^{90}$Sr intake function for Muslyumovo was reconstructed using a combination of $^{90}$Sr measurements in teeth, measured $^{90}$Sr concentrations in water and milk, and estimates of water and milk consumption rates for adults and children. For Metlino, data on the total-beta activity in excreta were also used [17]. The estimates of settlement-average-to-reference-village intake ratios ($f_{L_i(y)}^{90Sr}$, Eq 1) and corresponding village-specific shared uncertainties were based on the normalized $^{90}$Sr-body burdens in residents who lived continuously in the settlement of interest from 1950 until at least 1953, a period during which 95% of total $^{90}$Sr intake occurred. The main factor that determines $f_{L_i(y)}^{90Sr}$ variability within a specific village is the residents' primary water supply. The ratios of $^{90}$Sr intakes for different age groups to that for adults living in the reference settlement ($\alpha^{90Sr}$, Eq 1) were estimated using age-dependence of water- and milk-consumption rates and measurements of the ratio of $^{90}$Sr-concentration in milk to that in river water. Note that for most radionuclides, children-to-adult intake ratios are the same as for $^{90}$Sr since the main source of the intakes was the river water. One exception is $^{137}$Cs, which was additionally ingested with milk [17]. The model parameters and their uncertainty structure are described in Table 1.

These estimates of the individual radioisotope-specific annual activity intake are converted to the isotope-specific internal doses in various organs. The estimated dose to organ $o$ associated with the first year of chronic intake, $y_0$, from a given radioisotope is denoted as $d_{iro}^{T_{int}}(y, \tau_i(y_0))$,, where $y$ is the year of interest. This dose can be computed with a single intake

**Table 1. Summary table of the Techa River Dosimetry System (TRDS) parameters and their uncertainty structure.** Distributions are shown in terms of parameter multipliers as follows: *Norm*[mean, st. dev]; *LogNorm*[geometric mean, geometric st. dev]; *Uniform*[min/mean-max/mean]; *LogUniform*[min/mean-max/mean]; *Custom*–empirical distribution.

| Parameter | Description | Multiplicative uncertainty effect distribution | | | | Data source of uncertainty estimates |
|---|---|---|---|---|---|---|
| | | Shared (lack of knowledge) | | Unshared (stochastic) | | |
| | | shape | Number of parameters per individual and realization | shape | Number of parameters per individual and realization | |
| **Common to internal and external exposure** | | | | | | |
| $G^{90Sr}(L_i(y))$ | Surface deposition of $^{90}$Sr (Bq m$^{-2}$) at location $L$ from fallout from the EURT | *Norm*[1,0.05] | 84 villages →84 parameters | *Norm*[1,0.2] | Limit on moves set to 20 → 20 parameters | Shared–URCRM database Unshared–expert estimate |
| **Internal exposure** | | | | | | |
| $I_i^{T,r}(L_i(y), \tau_i(y))$ | Individual, $i$, intake from the Techa River (Bq) (function of age $\tau$ at year $y$) | | | | | |
| $I_y^{90Sr}$ | Annual $^{90}$Sr intake for adults of the reference settlements (reference $^{90}$Sr intake) | Within village or household Custom | 42 settlements → 42 parameters | *Norm*[1,0.25] | 53 time slice values → 53 parameters | Shared–from measurements of $^{90}$Sr body-burden Unshared–from measurement errors |
| $f_{L_i(y)}^{90Sr}$ | Ratio of $^{90}$Sr intake for location ($L$) to reference $^{90}$Sr intake | Considered in $I_y^{90Sr}$ | | | | Shared–from village-specific measurements of $^{90}$Sr body-burden if no individual |
| $\xi_i = \begin{cases} 1 \\ IMR_i \\ HSR_i \end{cases}$ | An individual modifier depending on availability and uncertainty of individual to model ratio ($IMR$) of $^{90}$Sr in the body or an average for household specific $IMR$s ($HSR$) | Considered in $I_y^{90Sr}$ | | | | $IMR_i$–unshared $HSR_i$–shared within household |
| $\alpha^{90Sr}(\tau_i(y))$ | Annual $^{90}$Sr intake for children relative to that for adults | within age group *Norm*[1,0.1] | Apply to one age to one year; after age 10, equal to adult → 11 parameters | *Norm*[1,0.2] | Apply to one age to one year; after age 10, equal to adult → 11 parameters | Shared–the error of the intake age dependence Unshared–expert estimate |
| $R_{L_i(y)}^{r,90Sr}$ | Annual-average location and time-specific ratio of radionuclide $r$-to-$^{90}$Sr intake | within village *LogNorm*[1,2] *Uniform*[0.5–1] | 11 nuclides × 42 villages → 462 parameters | | | Combined due to: global fallouts and radionuclide transport model [8] |
| $I_i^{E,r}$ | Annual radionuclide-specific intake function per unit of $^{90}$Sr surface deposition in EURT | within village *LogNorm*[1,3] | 6 nuclides x 84 villages → 504 parameters | *LogNorm*[1,2] | Limit on moves set to 20 → 20 parameters | URCRM database Shared–from errors of village-specific contamination Unshared–from studies of dietary intakes |
| $E_r(y, \tau_i(y))$ | Conversion factor for $^{90}$Sr surface deposition (Bq m$^{-2}$) to annual intake (Bq) for radioisotope $r$. Depend on age at time of intake and year after deposition. | Considered in $I_i^{E,r}$ | | | | Shared–from errors of age-dependence of dietary intakes |
| $DF_{ro}(t - y_0, \tau_i(y_0))$ | Conversion factor (Gy Bq$^{-1}$) for dose accumulated in organ $o$ in year $Y - y$ from intake of radionuclide r (function of age at intake) | within age with autocorrelation *Norm*[1,0.1] | 11 nuclides x 23 organs → 253 parameters | $^{90}$Sr -*LogNorm* [1,1.25] Other radionuclides -*LogNorm*[1,2] | 11 nuclides x 23 organs → 253 parameters | Shared–expert estimate Unshared–based on individual variability of skeleton mass (for $^{90}$Sr) or whole-body mass from literature data and URCRM database |
| **External exposure** | | | | | | |
| $A_o(\tau_i(y))$ | Conversion factor from absorbed dose in air to absorbed dose in organ $o$ (function of age at time of exposure) | within age with autocorrelation *Uniform*[0.9–1.1] | 23 organs → 23 parameters | *Uniform*[0.9–1.1] | 1 | Expert estimates |
| $D_{Riv}(y, L_i(y))$ | Annual absorbed dose in air on the Techa River shoreline at location $L$ (Gy year$^{-1}$) | within village with autocorrelation *Norm*[1,0.1] | 42 villages → 42 parameters | | | Expert estimates |
| $D^{90Sr}(L(y))$ | Normalized dose rate in air outdoors in time $y$ (Gy year$^{-1}$ per Bq m$^{-2}$) from EURT fallout | within village *Uniform*[0.9,1.1] | 15 month → 15 parameters | | | Expert estimates |

*(Continued)*

**Table 1.** (Continued)

| Parameter | Description | Multiplicative uncertainty effect distribution | | | | Data source of uncertainty estimates |
|---|---|---|---|---|---|---|
| | | Shared (lack of knowledge) | | Unshared (stochastic) | | |
| | | shape | Number of parameters per individual and realization | shape | Number of parameters per individual and realization | |
| $R_{\frac{out}{Riv}}(L_i(y))$ | Bank to residence ratio (function of distance of individual's home from river) | | | $LogUniform$[min-max]- village specific parameters | 42 villages → 42 parameters | Information about village-specific locations of households |
| $R_{\frac{in}{out}}(L_i(y))$ | Indoor/Outdoor ratio (function of building type) | | | $Uniform$[0.28, 1.72] | Limit on moves set to 20 → 20 parameters | Expert estimates |
| $T_1(\tau_i(y)), T_2(\tau_i(y))$ and $T_3(\tau_i(y))$ | Time fraction (annual-average) spent on the riverbank, outdoors and indoors, respectively | | | $LogNorm$[1, 2.7] | $T_1$ and $T_3$ are considered as independent → 2 parameters | Surveys-based estimates and expert estimates |

approach using the following expression (Eq 2):

$$d_{iro}^{T_{int}}(y - y_0, \tau_i(y_0)) = I_i^r(L_i(y_0), \tau_i(y_0)) \times DF_{ro}(y - y_0, \tau_i(y_0)) \tag{2}$$

where $DF_{ro}(y - y_0, \tau_i(y_0))$ is a dose coefficient to convert the radionuclide activity ingested in year $y_0$ to organ dose accumulated by the year of interest, y, (Bq to Gy). The dose factors are developed using biokinetic and dosimetric models. Accounting for chronic intake, the total internal isotope-specific dose to organ $o$ is (Eq 3)

$$d_{iro}^{T_{int,tot}}(y) = \sum_{t=y_0}^{y} d_{iro}^{T_{int}}(y - t, \tau_i(t)). \tag{3}$$

The total Techa internal dose to an organ in year y summed over all radionuclides can be written as (Eq 4)

$$D_{io}^{T_{int}} = \sum_r d_{iro}^{T_{int},tot} \tag{4}$$

**Techa River external dose.** The estimates of annual external dose rates from environmental exposure were based on a combination of archive data on dose rate and radionuclide activity measurements and the Techa River models [7, 8].

External doses for people living near the Techa River depend on estimates of the location-dependent annual dose rates in air at the riverbank, $D^{Riv}(y, L_i(y))$ modified by residence location within the village and individual behavioral patterns. The annual Techa external dose for organ $o$ for the age at time of exposure, $\tau_i(y)$, can be written as Eq (5):

$$d_{i,o}^{T_{ext}}(y) = A_o(\tau_i(y)) \times D_{Riv}(y, L_i(y))$$
$$\times \left[ T_1(\tau_i(y)) + R_{\frac{out}{Riv}}(L_i(y)) \times \left[ T_2(\tau_i(y)) + R_{\frac{in}{out}} \times T_3(\tau_i(y)) \right] \right] \tag{5}$$

where:

$D_{Riv}(y, L_i(y))$ is the absorbed dose in air near the river shoreline at location L in the year $y$;

$A_o(\tau_i(y))$ is the factor used to convert absorbed dose in air to organ dose;

$T_1(\tau_i(y))$ is the age-specific fraction of the year spent on the banks of the Techa;

$T_2(\tau_i(y))$ is the fraction of the year spent outside the house and not on the river bank;

$T_3(\tau_i(y))$ is the fraction of the year spent inside the house;

$R_{\frac{out}{Riv}}(L_i(y))$ is the residence to riverbank dose ratio at location L(y);

$R_{\frac{in}{out}}$ is the ratio of the dose rate inside the house to that outside the house.

The total external dose to a specific organ arising from Techa River exposures is given by Eq (6).

$$D_{io}^{T_{ext}} = \sum\nolimits_{t=y_o}^{y} d_{io}^{T_{ext}}(t) \tag{6}$$

## EURT doses

The doses received by EURT residents arose from exposure to $^{144}$Ce, $^{95}$Zr, $^{95}$Nb, $^{90}$Sr and $^{106}$Ru. Internal exposure was from the ingestion of contaminated milk and foodstuff; external exposure was due to surface deposition of these radionuclides, considered to be associated with $^{90}$Sr deposition. Surface contamination of soil with $^{90}$Sr was estimated based on the results of measurements of the dose rate of gamma radiation and the specific activity of radionuclides in environmental objects, which were performed immediately after the incident. Information on food contamination was also available and has been used for internal dose reconstruction.

**EURT internal dose.**   As with the Techa internal doses, the TRDS system provides separate internal organ dose estimates for each radioisotope that are then summed to provide time-dependent estimates of the annual organ doses. The expression for the annual radioisotope intake is (Eq 7)

$$I_i^{E,r} = G^{90_{Sr}}(L_i(y)) \times E_r(y, \tau_i(y)), \tag{7}$$

where:

$G^{90_{Sr}}(L_i(y))$ is the- surface $^{90}$Sr contamination of soil in the location $L_i(y)$ in year y;

$E_r(y,\tau_i(y))$ is a conversion factor from $^{90}$Sr *per Bq/m$^2$* to annual intake (Bq) for radioisotope *r*.

The estimated total internal organ dose in year *y* arising from chronic intakes since $y_0$ is computed as (Eq 8):

$$D_{io}^{E_{int}}(y) = \sum_r \sum_{t=y_0}^{y} \left[ I_i^{E,r}(L_i(y_0), \tau_i(y_0)) \times DF_{ro}(t - y_0, \tau_i(y_0)) \right], \tag{8}$$

where, as with the Techa internal doses, $DF_{ro}(t - y_0,\tau_i(y_0))$ is a dose factor that accounts for bio-kinetic and dosimetric effects to convert ingested radionuclide activity to organ dose. This dose factor is a function of age at intake and the time since the activity of interest was received.

**EURT external dose.**   The EURT external dose depends on individual residence history and behavior patterns (time spent inside the house). This annual dose was calculated as (Eq 9)

$$d_{io}^{E_{ext}}(y) = G^{90_{Sr}}(L_i(y)) \times A_0 \times D^{90_{Sr}}(L(y)) \times \left[ (1 - T_3) + T_3 \times R_{\frac{in}{out}} \right], \tag{9}$$

where

$D^{90_{Sr}}(L(y))$ is the normalized absorbed EURT fallout dose rate in air at location $L(y)$;

$T_3$ is the proportion of time spent inside house (this parameter has the same value as for the Techa River);

$R_{\frac{in}{out}}$ is the ratio of the dose rate inside the house to that outside the house.

## Monte Carlo dose realizations with shared and unshared uncertainties

The parameters of Eqs (1–9) are defined and elaborated in the frame of long-term study and summarized in [14, 18]. Time and settlement-specific parameters are organized in the

databases. Other parameters were characterized by distribution functions. The TRDS-2016MC databases include information on time-dependent radionuclide intakes and absorbed doses in air for 41 settlements located along the Techa River downstream from the site of radioactive releases to the mouth of the river, and 83 settlements located in the EURT area with initial deposition of $^{90}$Sr from 3.7 to 17800 kBq m$^{-2}$. The data for eight radionuclides ($^{89}$Sr, $^{90}$Sr, $^{95}$Zr, $^{95}$Nb, $^{103}$Ru, $^{106}$Ru, $^{137}$Cs, and $^{144}$Ce) are considered for both internal and external exposure. Each radionuclide has a separate table of age-dependent dose coefficients providing absorbed dose in organ *o* per unit intake. Table 1 presents the general description of the parameters and their uncertainty distribution. The characteristics of the distributions are presented in terms of multipliers on the arithmetic mean (equal to that in the deterministic version of TRDS). Details of data sources for parameter uncertainty distributions are presented in S1 File. These distributions reflected our understanding of the nature of the uncertainties as the system was developed.

As seen in Table 1, the behavioral parameters ($T_1, T_2$ and $T_3$) and the uncertainty in the residence type and location parameters ($R_{\frac{out}{Riv}}(L_i(y))$ and $R_{\frac{in}{out}}$) were considered to be purely unshared uncertainties. The location-dependent annual dose rate in air on the banks for the Techa ($D_{Riv}(y, L_i(y))$) and in the EURT ($D^{90Sr}(L(y))$) were assumed to be shared uncertainties with no unshared uncertainties. The remaining model parameters were assumed to have both shared and unshared errors. The uncertainty in the intake parameters has a major effect on both the shared and unshared errors in the individual annual dose estimates.

The TRDS-2016MC code provides multiple realizations of individual annual internal and external doses from both the Techa and 1957 accident exposures for each member of the Techa River /EURT combined cohort. This was accomplished using a two-dimensional Monte Carlo method (2DMC) [15] as follows.

In the first step of the 2DMC process, a set of multiplicative shared uncertainty factors was sampled from the parameter-specific shared uncertainty distributions for each realization and used to compute a set of model parameters, which were stored in a database with one record per realization. This sampling was done using a Latin hypercube sampling [19, 20] procedure that requires at least one realization per parameter in the dosimetry system. As seen in Table 1, the system involves 1,436 shared and 422 unshared parameters. Additionally, age dependent $^{137}$Cs intakes were considered separately for contamination and consumption of water and milk. We carried out 1,500 realizations for each annual dose from the start of follow-up to the earliest of the date of death, last known residence in the Chelyabinsk or Kurgan Oblast, or the end of 2016. In the second step of the 2DMC process, the realization-specific model parameters for each person were perturbed by Monte-Carlo sampling of the unshared multiplicative error factors for each parameter.

Using the stored realization-specific shared parameter estimates, dose realizations were carried out separately for four exposure components: Techa River external exposures; Techa River internal exposures; EURT external exposures and EURT internal exposures. For each exposure component individual annual doses were provided for 23 organs. These are: esophagus; stomach; small intestine; colon; rectum; lungs; breast; active bone marrow; bone surface; thyroid; bladder; liver; spleen; kidneys; pancreas; adrenals; thymus; uterus; testes; ovaries; brain; muscle and skin. Each realization included annual organ dose estimates for 48,036 members of the combined cohort with up to 67 years of follow-up for each person. Individual arithmetic and geometric means as well as corresponding CVs and GSDs were calculated over the 1,500 dose realizations for each person. Population arithmetic and geometric means and CVs and GSDs were computed using the individual arithmetic and geometric mean values.

### Ethics statement

This work is part of an ongoing study of radiation-exposed populations in the Russian Southern Urals. Since the 1990's, both the dosimetric work and the related epidemiological studies of the data collection and analysis methods used for the study of the Techa River and EURT cohorts have been regularly reviewed and approved by the Expert Commission that serves as the Institutional Review Board (IRB) of the Urals Research Center for Radiation Medicine (URCRM). The URCRM IRB has determined that individual consent was not necessary for members of the study populations. In addition, the project was routinely reviewed by the U.S. Department of Energy's Scientific Review Group.

## Results

In this section, we present summary information on external and internal exposures arising from Techa River and EURT exposures in the combined TRC and EURTC cohorts. These summaries include information on:

- the distribution of the within-individual variation in the realized dose components,

- the population dose distributions for the individual dose components and total doses, and

- descriptive information illustrating the nature of the impact of the shared uncertainty

TRDS-2016MC dose estimates were computed for 48,036 people, 47,950 of which were later included in the combined cohort (29,709 TRC members and 19,839 EURTC members with 1,598 people who were in both the TRC and EURTC). There were 29,859 people who received Techa River exposures and 23,710 with EURT exposures. Among these people, 5,534 people had exposures from both sources. In addition to the 1,598 people in the combined cohort who were in both the TRC and EURTC cohorts, there were 116 EURTC members with some Techa River exposure and 3,807 TRC members with some EURT exposure. Among the 86 people with TRDS2016 doses who were not included in the combined cohort, 52 had only Techa River exposures, 21 only EURT exposures and 13 had both Techa River and EURT exposures.

### External doses

Table 2 provides summary information about the population distribution of the total individual arithmetic and geometric mean doses for external TR and EURT stomach and marrow doses. These arithmetic and geometric mean doses were computed using the 1500 dose realizations for each person in the cohort. The table also provides information on the variability of the individual realizations in terms of the distribution of the within-person coefficient of variation and the GSD for total dose.

In the combined cohort, the mean Techa River external doses range up to 678 mGy for the stomach and 765 mGy for the marrow with means (medians) of 26 (2.5) and 29 (3), respectively. While the maximum external dose estimates from EURT exposures are about one-seventh those arising from TR external exposures, the EURT population mean and median stomach and marrow doses are only about one third of the corresponding TR doses. The dose difference between organs is due to the *weak* photon-energy dependence of the absorbed-dose-in-air to absorbed-dose-in-organ $o$, $A_o$ (Eq 1). The external exposure in the Urals region was mainly due to environmentally distributed $^{137}$Cs/$^{137m}$Ba, $^{95}$Zr, $^{95}$Nb, $^{144}$Ce/$^{144}$Pr/$^{144m}$Pr and $^{106}$Ru/$^{106}$Rh which are gamma-emitters with photon energies within the range of 81–1050 keV. However, there is a large plateau in the energy dependence of $A_0$ between about 80 and 1300 keV. Therefore, the mean value of $A_0$ for these gamma-emitters was fixed to be equal to

Table 2. Description of external dose realization descriptive statistics.

| Summary Statistic | Mean | CV* | Percentiles | | | | | |
|---|---|---|---|---|---|---|---|---|
| | | | 5 | 25 | Median | 75 | 95 | Max |
| | | | Stomach Dose | | | | | |
| | | | Techa River (N = 29,859) | | | | | |
| Population mean (mGy)* | 26 | 2.83 | 0.09 | 0.8 | 2.5 | 13 | 151 | 678 |
| Population Geometric Mean (mGy)* | 18 | 2.94 | 0.05 | 0.5 | 1.6 | 8 | 100 | 550 |
| Individual Coefficient of Variation** | 1.13 | 0.14 | 0.89 | 1.01 | 1.1 | 1.25 | 1.36 | 2.27 |
| Individual Geometric Standard Deviation** | 2.56 | 0.11 | 2.12 | 2.38 | 2.61 | 2.75 | 2.91 | 5.31 |
| | | | EURT (N = 23,710) | | | | | |
| Population mean (mGy)* | 7 | 2.32 | 0.02 | 0.31 | 1.2 | 6.57 | 58.07 | 88 |
| Population Geometric Mean (mGy)* | 5 | 2.32 | 0.01 | 0.23 | 1.0 | 4.99 | 41.60 | 64 |
| Individual Coefficient of Variation** | 0.97 | 0.14 | 0.78 | 0.83 | 1.0 | 1.06 | 1.15 | 2.65 |
| Individual Geometric Standard Deviation** | 2.17 | 0.04 | 2.04 | 2.07 | 2.22 | 2.24 | 2.26 | 2.30 |
| | | | Marrow Dose | | | | | |
| | | | Techa River (N = 29,859) | | | | | |
| Population mean (mGy)* | 29 | 2.83 | 0.10 | 0.9 | 3.0 | 14 | 169 | 765 |
| Population Geometric Mean (mGy)* | 21 | 2.94 | 0.06 | 0.6 | 2.0 | 9 | 112 | 622 |
| Individual Coefficient of Variation** | 1.13 | 0.14 | 0.89 | 1.0 | 1.14 | 1.25 | 1.36 | 2.28 |
| Individual Geometric Standard Deviation** | 2.56 | 0.11 | 2.12 | 2.38 | 2.61 | 2.75 | 2.91 | 5.32 |
| | | | EURT (N = 23,710) | | | | | |
| Population mean (mGy)* | 8 | 2.32 | 0.02 | 0.3 | 1.4 | 7 | 65 | 100 |
| Population Geometric Mean (mGy)* | 6 | 2.31 | 0.01 | 0.3 | 1.1 | 6 | 46 | 72 |
| Individual Coefficient of Variation** | 0.97 | 0.14 | 0.78 | 0.83 | 1.0 | 1.06 | 1.15 | 2.51 |
| Individual Geometric Standard Deviation** | 2.17 | 0.04 | 2.04 | 2.07 | 2.22 | 2.24 | 2.26 | 2.30 |

* For the population mean and geometric mean rows the values shown are the coefficient of variation (CV) of the arithmetic and geometric mean values over the population.

** For the individual CV and geometric standard deviation (GSD) rows these are the average of the CV and GSD values within realization

those typical of $^{137}$Cs/$^{137m}$Ba [21]. In other words, the width of individual dose distribution is not organ specific. Therefore, the mean GSD and CV are equal for marrow and stomach. On average, the uncertainties of cumulative EURT external doses are 15% lower than those for the Techa River doses. This is mainly due to the shorter period of exposure in the EURT, where the short-lived radionuclides (all radionuclides except $^{137}$Cs) were the main contributors. Distributions of individual external dose uncertainties in terms of GSD are shown in Fig 1A and 1B for those who were exposed from the Techa River and in the EURT, respectively. As one would expect the CV of the population means (in the CV column of the mean dose rows in the table) are considerably greater than the individual CV and GSD values.

The Techa River GSD distribution has five modes indicated by the numbered arrows in Fig 1A. The first mode corresponds to GSD = 1.91 (CV ∼ 75%), the second–GSD = 2.4 (CV ∼ 110%), the third–GSD = 2.65 (CV ∼ 120%), the fourth–GSD = 2.9 (CV ∼ 130%) and the fifth–GSD = 3.3 (CV ∼ 140%). The cumulative dose uncertainties largely depend on residence in the year of initial exposure, the duration of residence in contaminated areas, and movements between settlements in contaminated areas.

For permanent residents (at least in the period from 1950 to 1956), the GSD modes are largely due to uncertainties of the outdoor-to-riverbank dose rate ratio parameter– $R_{\frac{out}{Riv}}(L_i(y))$. TRDS-2016MC applies a subdivision of 9 Upper Techa settlements into 47 territories with homogenous conditions of external exposure (clusters) based on information available. The

mean of the cluster-specific $R_{\frac{out}{Riv}}(L_i(y))$ and the range of possible values were considered as unshared parameters when simulating the variability of the rates within a cluster (S1 File). For the Middle and Lower Techa villages the settlement-average $R_{\frac{out}{Riv}}(L_i(y))$ value is used. The lowest values of $R_{\frac{out}{Riv}}(L_i(y))$ uncertainty were typically for individuals with known household location within a village. The houses in three small settlements from the lower reaches (Vetrodujka, Zamanikha and Russkaya Techa) were parallel to the river so that the range of possible values of $R_{\frac{out}{Riv}}(L_i(y))$ was small and, accordingly, the uncertainties of this parameter were comparable to that for the village-specific clusters in the Upper Techa. Generally, the uncertainty of $R_{\frac{out}{Riv}}(L_i(y))$ values in the village-specific clusters was proportional to the mean value of the parameter. For example, people in GSD mode 1 were typically residents of Metlino (7 km from the release site) for whom $R_{out/Riv,L} > 0.03$; the second mode corresponds to $0.006 < R_{out/Riv,L} < 0.03$; the third one to $0.006 > R_{out/Riv,L}$.

Modes 4 and 5 include people with doses calculated based on village-average $R_{\frac{out}{Riv}}(L_i(y))$. The difference between the modes of these two groups depends primarily on the residence history, age at time of exposure (correlated to time spent at shoreline), and distance from the release site. The maximum uncertainties are typical of the smallest doses. For example, cumulative doses in the Lower Techa region (>200 km from site of releases) are almost always less than 4 mGy; while the GSDs are quite large (>3).

The distribution of individual GSDs of cumulative EURT organ doses has two pronounced modes (Fig 1B), with GSDs of 2.06 (CV ~ 78%) and 2.25 (CV ~ 105%), respectively. The two groups largely reflect age at exposure during the year, which is correlated with time spent indoors. The first group consists of those whose age were less than 7 or older than 60 at the time of the accident in 1957 (the outdoor time fraction <30%). The second mode reflects people of ages between 7 and 59. The variability within the modes reflects uncertainties induced by individual residence histories.

Fig 2 presents examples of the mean cumulative stomach dose realizations for two cohort members. Person 1 was born in Metlino (7 km from the site of releases) and lived at a known location in the village ($R_{out/Riv,L} > 0.03$) prior to evacuation to ONIS in 1956. The stomach dose accumulated between 1950 and 1956 was 0.59 Gy (CV = 78%); GSD = 1.92 (corresponding to the first mode in Fig 1A). The EURT-related contamination levels in ONIS in 1957 were 56 kBq m$^{-2}$. The individual had lived in ONIS from the time of the accident until 1960. The mean stomach dose accumulated from EURT exposures is equal to 0.002 Gy; GSD = 2.24 (corresponding to the second mode in Fig 1B).

Person 2 is an individual born in 1943, initially exposed in the EURT (Russkaya Karabolka, 2427 kBq m$^{-2}$) until evacuation to Russkaya Techa, on the Techa River 137 km from the site of releases, in 1958. The mean cumulative EURT stomach dose was 0.04 Gy (CV = 82%); GSD = 2.07 (the first mode in Fig 1B). Between 1958 and 1970 this person received an additional 0.002 Gy with a GSD = 2.58 (the third mode in Fig 1A).

For Person 1, the total mean cumulative dose was dominated by the mean cumulative TR dose which exceeded the mean cumulative EURT dose by two orders of magnitude and essentially determined the total probability density function (Fig 2B). In the second case, the mean EURT dose was, on average, an order of magnitude greater than the TR dose with a slight realization overlap (< 5%) with those for the Techa River. The total cumulative dose distribution of total doses was largely determined by the EURT dose. The lognormal-like asymmetry of these dose distributions is typical of those for most exposures. Smoothing of histograms of cumulative dose densities were computed using non-parametric kernel density estimation

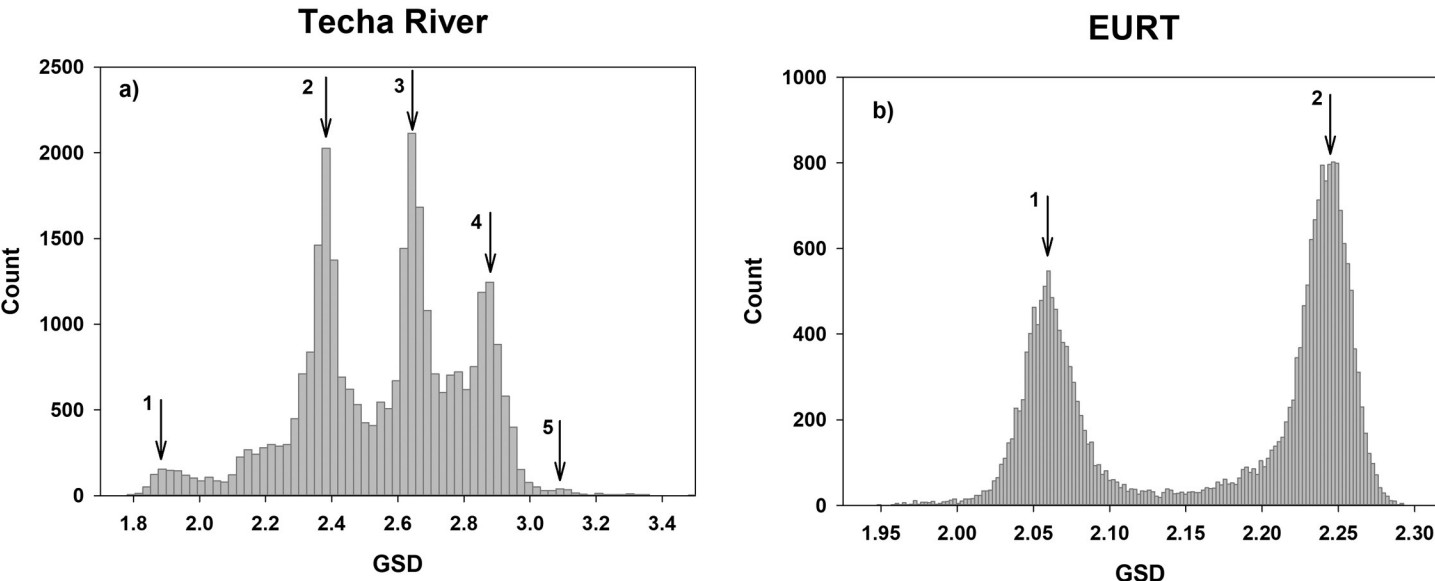

**Fig 1.** Density of uncertainties in individual total cumulative dose from Techa River (a) and EURT (b) external exposures. The numbered arrows indicate local modes that are referenced in the text.

[22]. The distributions of the individual total external stomach dose realizations for each of these people were well-described as lognormal.

### Internal doses

In contrast to external exposure, the main contributors of internal dose in bone marrow and stomach or other soft tissues differ markedly. Soft tissues were primarily exposed due to circulating radionuclides during the intake period. The marrow was chronically exposed to the activity of long-lived bone-seeking $^{90}$Sr throughout their lives in addition to the circulating radionuclides. For $^{90}$Sr, both the biokinetic and dosimetric models are quite different for marrow and soft tissue doses. Therefore, uncertainties of marrow and soft tissue dose coefficients ($DF_{ro}(y - y_0, \tau_i(y_0))$ in Eq.2) comprised of dosimetric and biokinetic models also differed. Table 3 summarizes the distribution of internal doses (from both the Techa River and EURT) and corresponding organ-specific uncertainties for stomach and marrow. As seen in Table 3, the mean total cumulative internal stomach doses from the Techa River were as large as 541 mGy with a mean (median) of 31 (18) mGy. Because of the chronic bone surface exposures, Techa River internal marrow doses were about 10 times the internal stomach doses with a mean (median) of 294 (174) mGy and a maximum of slightly more than 6.7 Gy. On average, EURT internal exposures are less than 10% of the corresponding Techa River exposures (Table 3).

For Techa River residents with $^{90}$Sr body burden measurements, it was possible to estimate a time-integrated individual-to-model-ratio ($IMR_i$) value that reflects the ratio of the individual body burden to the village-average age- and gender-specific body burden predicted by the strontium biokinetic model [23]. When IMRs were available within a household, the average of the household-member-specific IMRs were used to define a *Household-specific-ratio* (*HSR*) that was then used to adjust the village-average intake for other cohort members who lived in the same household. This approach made it possible to avoid explicit simulation of the various

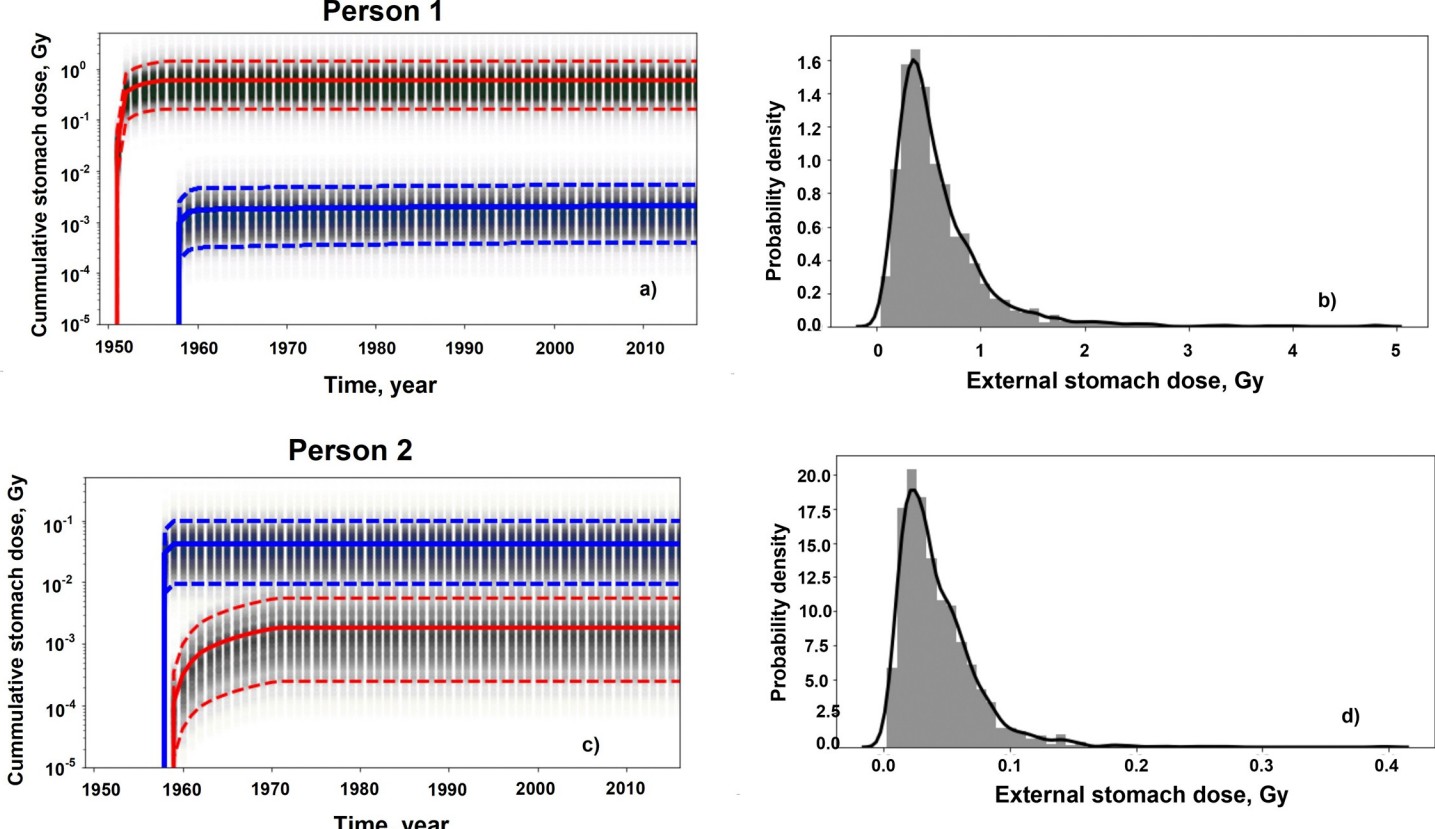

**Fig 2. Examples of external stomach dose realizations for two cohort members.** The plots in the top row (a and c) summarize the distribution of the time-dependent annual cumulative dose realizations from the Techa River (red curves) and EURT (blue curves). In these plots, solid lines represent the arithmetic means and dashed lines indicate the 5th and 95th percentiles of the annual total cumulative dose realizations. The vertical bars for each year reflect the probability density of the realizations in that year. The plots in the bottom row (b and d) show the density of the total external doses at the end of the follow-up with a smoothed estimate (solid black lines).

components of drinking water source, diet, uptake, or metabolism that affect $I_i^{T,r}(L_i(y), \tau_i(y))$ and greatly simplifies the uncertainty description (which becomes mainly unshared).

Distribution of village-average $I^*_{y,r,L}$ estimates is characterized by 90% CI for the village mean intakes that ranged from 1.88 to 5.28. *IMR* or *HSR*-based dose individualization is available for about 6,500 and 8,500 individuals, respectively. The use of the *IMR*- or *HSR*-based individual modifiers reduces the uncertainty in $I^*_{y,r,L}$ by about 40%.

The contribution of exposures from non-$^{90}$Sr radionuclides ($^{89}$Sr, $^{141,144}$Ce, $^{95}$Zr, $^{95}$Nb, $^{137}$Cs, $^{103,106}$Ru) to internal exposure is an additional source of internal dose uncertainty. This contribution was described using time-dependent, radionuclide-specific ratios of nuclide-to-$^{90}$Sr intake ($R_{L_i(y)}^{r:90Sr}$). Radionuclide intake for the Techa River settlements was primarily from drinking water. Radionuclide-specific ratios at different locations along the stream were estimated based on models for radionuclide transport in the river [8]. The uncertainties of $R_{L_i(y)}^{r:90Sr}$ are shared within villages and correspond to uncertainties of the model prediction. Intake of $^{137}$Cs from cow's milk was also estimated for several villages [17]. At distances 7–48 km from the site of the releases, the floodplain was waterlogged and was not used as a source of cow-forage. However, beyond 48 km from Mayak, the floodplain was extensively used as a source of cow forage and cow's milk was the main source of $^{137}$Cs in local diet. The $^{137}$Cs intake with

**Table 3. Techa river and EURT internal dose realization descriptive statistics.**

| Summary Statistic | Mean | CV* | Percentiles | | | | | |
|---|---|---|---|---|---|---|---|---|
| | | | 5 | 25 | Median | 75 | 95 | Max |
| **Stomach Dose** | | | | | | | | |
| **Techa River (N = 29,859)** | | | | | | | | |
| Population Arithmetic mean (mGy)* | 31 | 1.32 | 0.2 | 5.8 | 18 | 38 | 104 | 541 |
| Population Geometric Mean (mGy) * | 24 | 1.38 | 0.15 | 4.0 | 12 | 30 | 85 | 463 |
| Individual Coefficient of Variation** | 0.98 | 0.31 | 0.6 | 0.7 | 0.9 | 1.16 | 1.56 | 4.14 |
| Individual geometric standard deviation** | 2.23 | 0.20 | 1.71 | 1.83 | 2.18 | 2.45 | 3.14 | 4.45 |
| **EURT (N = 23,710)** | | | | | | | | |
| Population Arithmetic mean (mGy)* | 2.6 | 3.10 | 0.002 | 0.07 | 0.3 | 1.70 | 13 | 79.83 |
| Population geometric mean (mGy) * | 1.2 | 2.93 | 0.001 | 0.04 | 0.15 | 0.8 | 6 | 32.15 |
| Individual coefficient of variation** | 2.02 | 0.28 | 1.48 | 1.71 | 1.90 | 2.17 | 2.89 | 17.21 |
| Individual geometric standard deviation** | 3.09 | 0.07 | 2.71 | 2.99 | 3.10 | 3.22 | 3.49 | 3.99 |
| **Marrow Dose** | | | | | | | | |
| **Techa River (N = 29,859)** | | | | | | | | |
| Population Arithmetic mean (mGy)* | 294 | 1.30 | 1.2 | 37 | 174 | 395 | 1011 | 6738 |
| Population geometric mean (mGy) * | 185 | 1.49 | 0.7 | 21 | 82 | 238 | 707 | 4971 |
| Individual coefficient of variation** | 1.43 | 0.32 | 0.82 | 1.0 | 1.50 | 1.81 | 2.13 | 3.32 |
| Individual geometric standard deviation** | 2.94 | 0.29 | 1.94 | 2.16 | 2.76 | 3.74 | 4.37 | 4.98 |
| **EURT (N = 23,710)** | | | | | | | | |
| Population Arithmetic mean (mGy)* | 18 | 1.91 | 0.21 | 2.1 | 5.8 | 16 | 100 | 298 |
| Population geometric mean (mGy) * | 8 | 1.94 | 0.09 | 0.9 | 2.6 | 7 | 46 | 125 |
| Individual coefficient of variation** | 2.06 | 0.21 | 1.58 | 1.82 | 2.0 | 2.20 | 2.78 | 8.50 |
| Individual geometric standard deviation** | 3.54 | 0.08 | 2.85 | 3.42 | 3.65 | 3.74 | 3.83 | 4.01 |

* The statistics in the population arithmetic and geometric mean rows are based in the values of the within- individual arithmetic and geometric means, which were computed using the 1500 realizations for each person

** The statistics on the individual coefficient of variation (CV) and geometric standard deviation(GSD) rows are based on the distribution of the within individual CVs and GSDs

cow's milk was assumed to have a small shared uncertainty and a larger unshared uncertainty component due to individual variations in milk consumption.

Fig 3 shows the densities of the individual GSD estimates for internal marrow and stomach dose from Techa River and EURT exposures.

For Techa marrow dose realizations (Fig 3A), two peaks are observed for GSD <2.3. This GSD range consists of marrow dose realizations for people with individualized assessments of $^{90}$Sr intakes. The *IMR* scaling factor uncertainty for an individual is reduced as the number of $^{90}$Sr body burden measurements increased. The first peak, GSD = 1.8 (CV ∼ 70%), corresponds to the doses for people with reliable multiple measurements of $^{90}$Sr body burden, mainly for the residents of the Upper Techa (<48 km) with negligible impact of $^{137}$Cs intakes from cow's milk or who resided in the contaminated areas for a relatively short period (<4 years). The second peak, GSD 2.2 (CV ∼ 100%), includes people with few individual measurements as well as people living more than 48 km from the release point with a reasonable number of reliable $^{90}$Sr measurements but large uncertainties in $^{137}$Cs intake. Together, these two peaks include data for about 10,000 people. The dose realization GSD for other cohort members are rather uniformly distributed. The largest GSDs correspond to people with mean total doses below 0.001 Gy. In contrast, the soft tissue doses are largely due to $^{137}$Cs exposures. Two

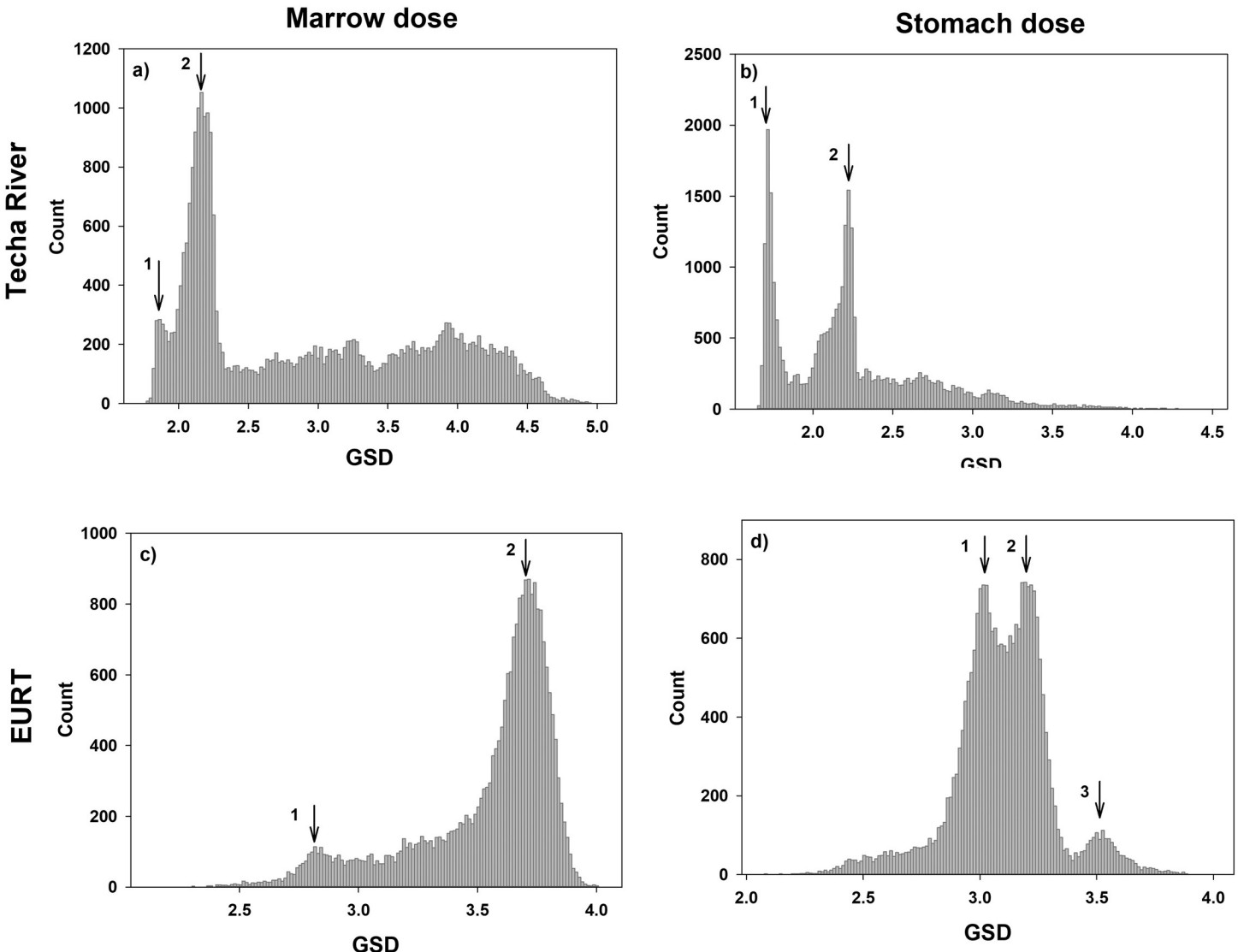

**Fig 3.** Distribution of individual internal dose uncertainties for active bone marrow and stomach exposure at the Techa River (a and b, respectively) as well as at EURT (c and d, respectively).

peaks (GSD = 1.7 (CV ∼ 64%) and GSD = 2.2 (CV ∼ 84%)) in Fig 3B correspond to individuals who permanently lived above and below 48 km from the release point.

For EURT, intake per unit surface deposition of $^{90}$Sr, $E_{r,y}$, was evaluated using a database of $^{90}$Sr measurements in foodstuffs and human body/bone/excreta [24]. The ratios of $^{90}$Sr content in children's diet to that for adults were calculated based on $^{90}$Sr contamination of local foodstuffs and age-related diet compositions. Uncertainty distributions for this parameter were developed to account for the uncertainty in radionuclide composition of the release, non-uniform contamination of village-related territories (which resulted in variability of harvest contamination), individual dietary variability, and some other factors. The composite of these factors is a lognormal distribution for a reference adult with GSD of about 2.7, of which a portion is shared and a portion unshared between members of the cohort. For non-strontium radionuclides, intakes were estimated based on data on total beta-activity in foodstuff adjusted

for radionuclide composition. In contrast to dose reconstruction for the Techa River, EURT doses were not individualized, and the primary source of individual-to-individual variation in cumulative dose uncertainties was related to residence history.

As for the Techa doses, bone seeking $^{90}$Sr had a marked influence on the marrow dose. The first peak in the EURT marrow GSD density (Fig 3C) corresponds to the data for people with short-term intakes, mainly residents of three settlements (Berdyanish, Satlykovo, and Galikaeva) located in the most contaminated EURT area (17,760 kBq m$^{-2}$ of $^{90}$Sr deposition), which were evacuated within 7–14 days after the accident. The average $^{90}$Sr exposure was almost 500 kBq of $^{90}$Sr per stay. The first peak (GSD = 2.8; CV $\sim$ 140%) is mainly defined by the uncertainty of $I_i^{E,r}$. The second peak (GSD = 3.74; CV $\sim$ 200%) is the modal value for all other EURT residents.

Non-strontium isotopes were the main source of soft tissue exposure in the initial period after the incident [24]. For example, for settlements evacuated during the first two weeks after the explosion, the isotope-to-strontium concentration ratios were considered as follows: $^{144}$Ce —121; $^{106}$Ru—7; $^{95}$Zr—34; $^{95}$Nb—57; $^{137}$Cs—0.64. Information on non-strontium-radionuclide contamination of food is limited, therefore, intake estimates for the early exposure periods were highly uncertain. As a result, the 3$^{rd}$ peak of stomach dose uncertainty distribution in Fig 3D (GSD = 3.59, CV $\sim$ 240%) corresponds to doses calculated for residents evacuated within two weeks of the explosion. The 1$^{st}$ and 2$^{nd}$ peaks in Fig 3D (GSD = 2.98, CV $\sim$ 190% and GSD = 3.24, CV $\sim$ 210%, respectively) are based on the much larger number of exposed people who were evacuated at later dates or were never evacuated. Clustering of these data depended mainly on the age at the initial intake, usually the time of the accident. Group 1 is dominated by individuals whose were infants (0–2 years old) in 1957 and for whom individual variability in diets does not play a big role in the structure of overall uncertainty.

Fig 4 presents information on internal marrow and stomach dose realizations for two people. The upper plots for each person illustrate the annual cumulative mean (solid curves) and the 5$^{th}$ and 95$^{th}$ percentiles (dashed curves) of the annual values. Red curves represent the Techa doses and blue curves the EURT doses. The density of the values for each year is indicated by the varying intensity of the shaded vertical bars.

Person 3 was a male born in 1926 who lived in the Techa River village of Nadirovo, 50 km from the site of releases from1950 until moving to the Techa river settlement of GRP in 1954. Between 1957 and 1960, this person lived in the EURT village of Taskino with a $^{90}$Sr surface contamination level of 11 kBq m$^{-2}$. Between 1976 and 1989, this person had 19 strontium body-burden measurements. As a result, the internal doses due to $^{90}$Sr (1.1 Gy and 0.33 Gy for total marrow and total stomach doses, respectively) were estimated quite precisely (CV $\sim$ 80%; GSD $\sim$ 2) for both marrow and stomach. Geometric means are 0.89 Gy and 0.26 Gy, respectively. The dose uncertainty for EURT exposure was much higher due to the high contribution of non-strontium isotopes (GSD $\sim$ 3.8). However, the cumulative doses from EURT exposures (<0.002 Gy) were 2 to 3 orders of magnitude lower than the Techa River doses (0.9 and 0.26 Gy for marrow and stomach, respectively).

In contrast, person 4 was a female born in 1948 who resided along the Techa River in Novoe Asanovo, 30 km from the site of releases from 1950 until she moved to ONIS in 1955. Following the 1957 accident, surface contamination levels in ONIS were estimated to be 56 kBq m$^{-2}$ of $^{90}$Sr surface contamination. The internal dose estimates for this person (1.05 Gy and 0.07 Gy for total marrow and total stomach doses, respectively) are highly uncertain (CVs are 160% and 135%, respectively) since they had no $^{90}$Sr measurements and the exposures were based on settlement-average values. Geometric means are 0.56 Gy and 0.04 Gy, respectively. The GSDs were calculated to be about 4 for total marrow dose and 3.3 for total stomach

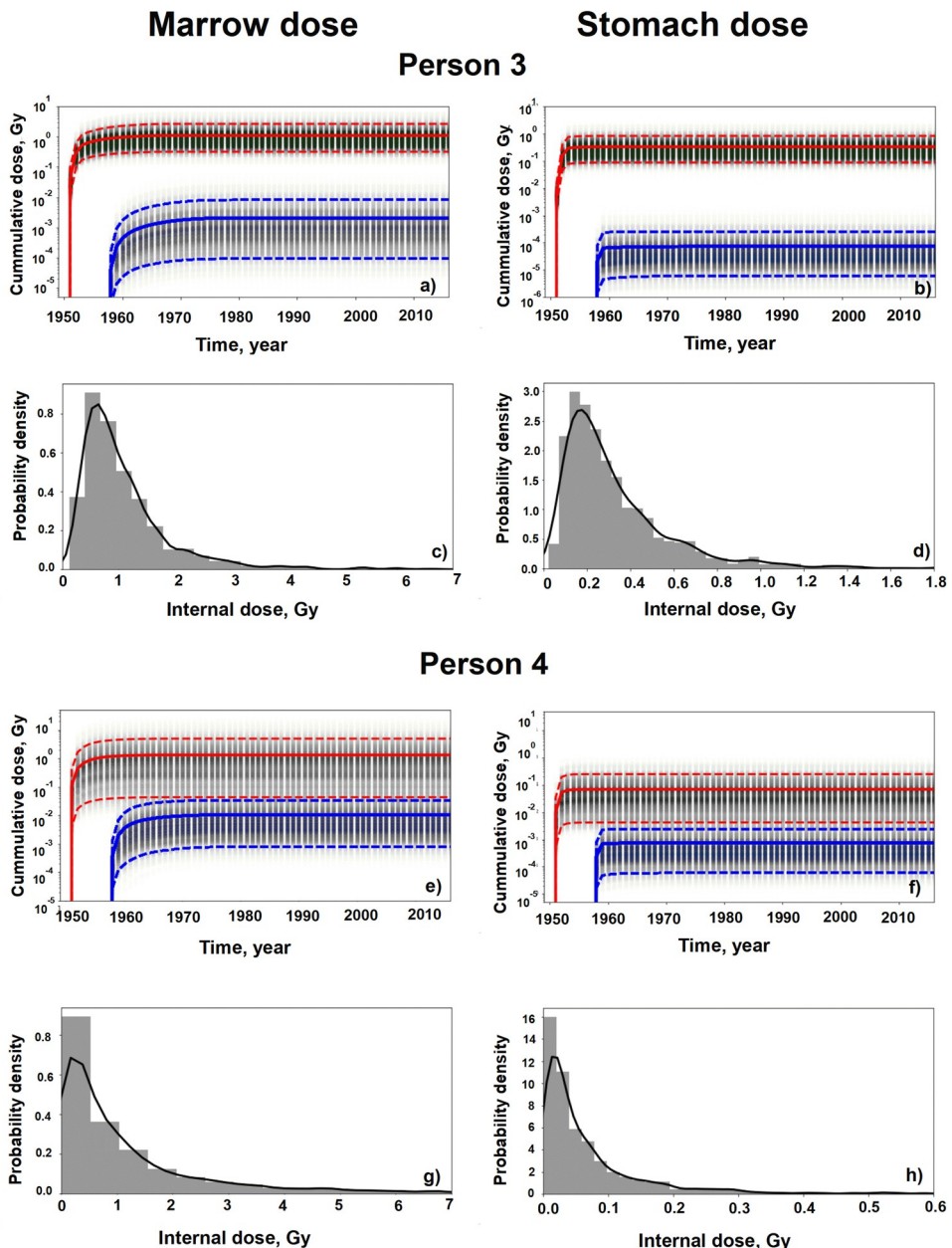

**Fig 4. Examples of internal marrow and stomach dose realizations for two cohort members.** For each person the plots in the top row summarize the distribution of the time-dependent annual cumulative dose realizations from the Techa River (red curves) and EURT (blue curves). In these plots solid lines represent the arithmetic means and dashed lines indicate the 5th and 95th percentiles of the annual total cumulative dose realizations. The vertical bars for each year reflect the probability density of the realizations in that year. The plots in the bottom row for each person show the density of the total internal dose realizations at the end of the follow-up with a smoothed estimate (solid black lines).

dose. The contribution to the doses arising from residence in the EURT is on average 2 orders of magnitude lower than that for the Techa River. However, in a small proportion of the realizations the EURT dose is slightly greater than the Techa River dose.

The plots in the bottom row for each person illustrate the cumulative dose realization densities at the end of follow-up. The solid lines are smoothed kernel density estimates computed

using the KDE non-parametric approach. In each of these cases, the densities suggest a skewed lognormal-like distribution.

## Total absorbed doses

Table 4 summarizes the population and individual uncertainties in individual total cumulative marrow and stomach dose for all members of the combined cohort. This is the sum of the annual internal and external doses arising from both Techa River and EURT exposures.

Typically, among cohort members with both TR and EURT exposures, doses from EURT exposure were lower than those from Techa River (as illustrated in Figs 2 and 4 for persons 1, 3, and 4). As a result, for people with both TR and EURT exposures the uncertainty of total dose was largely determined by the uncertainty of the Techa River dose components.

Radiation exposure in these environmentally exposed Urals' populations resulted in non-uniform dose accumulation within the body. Bone-seeking $^{90}$Sr results in marrow doses that are at least an order of magnitude higher than soft tissue doses. For most cohort members, internal exposure was the largest component of the total marrow dose, comprising more than half of the cumulative marrow dose for 92% of cohort members. The cohort-average contribution of internal dose to total marrow dose was 84%. Therefore, the total marrow dose GSD density (Fig 5A) has a narrow peak at 2.2 and long tail up to 5 like that seen for the internal Techa River dose component (Fig 3A). External dose uncertainty also contributed to the marrow dose uncertainty. This uncertainty is reflected in the mode around a GSD near 2.5 – reflecting the modal GSDs around 2.4 and 2.7 seen in the Techa River external dose estimates (Fig 1A). In general, people for whom the GSD is less than 3 are cohort members with individualized external and/or internal doses. The larger marrow GSDs were seen for people whose doses were based on group average exposure estimates, which result in larger shared uncertainties.

Averaging over all cohort members, 52% of the stomach dose was from internal exposure. In particular, internal exposure accounts for less than half of the total stomach dose for roughly 50% of the survivors. Nevertheless, the pronounced multimodality seen for TR external dose (Fig 1A) is less apparent when looking at total stomach dose (Fig 5B). Uncertainty of total stomach dose clusters in 2 peaks and a long tail. The first peak, around a GSD of 1.8, is typical of the Techa River residents with individualized estimates for both the external and internal dose components. The peak around a GSD of 2.3 corresponds to doses for people with limited individualization, while the long tail reflects the uncertainty in people for whom there was no individualization.

We examined the nature of the cumulative distributions of the realizations for individual cohort members by determining which of several candidate distributions best described the empirical distribution of total dose for that person. The goodness of fit was assessed using one-sample Kolmogorov-Smirnov (K-S) tests [25] for each member of the combined (EURT + the Techa River) cohort. The best fitting distribution was taken to be the candidate distribution with the largest P-value greater than 0.05. If the P-values for each of candidate distributions was less than 0.05, we indicate that none of the distributions fit well. Table 5 shows the proportion of the best fitting cumulative distributions for individual dose realizations by exposure type and source components for the total dose.

The distributions of dose estimates over dose realization for individual cohort members were highly skewed to the right, which is to be expected given the nature and complexity of the uncertainty factor distributions. For both Techa River internal and external total cumulative doses, the dose realization distributions were well described by lognormal distributions for more than two-thirds of the cohort members. For doses attributed to EURT internal

**Table 4. Summary information on the population distribution of individual* arithmetic (M) mean and coefficient of variation (CV) and individual geometric (GM) and geometric standard deviation (GSD) values for total active marrow and stomach dose estimates for the 48,036 people in the combined TRC and EURTC cohorts.**

| Population† distribution parameters | M, Gy | CV | GM, Gy | GSD |
|---|---|---|---|---|
| *Active Marrow* | | | | |
| Arithmetic mean | 0.21 | 1.56 | 0.21 | 2.93 |
| Standard deviation | 0.36 | 0.47 | 0.30 | 0.81 |
| min | 0.00002 | 0.67 | 0.00001 | 1.80 |
| 5% | 0.001 | 0.89 | 0.001 | 2.02 |
| 25% | 0.009 | 1.15 | 0.03 | 2.19 |
| 50% | 0.06 | 1.61 | 0.09 | 2.70 |
| 75% | 0.26 | 1.85 | 0.27 | 3.67 |
| 95% | 0.93 | 2.24 | 0.78 | 4.34 |
| max | 6.75 | 7.75 | 4.98 | 4.95 |
| *Stomach* | | | | |
| Arithmetic mean | 0.04 | 1.10 | 0.03 | 2.32 |
| Standard deviation | 0.08 | 0.25 | 0.06 | 0.33 |
| min | 0.000002 | 0.58 | 0.000001 | 1.68 |
| 5% | 0.0002 | 0.68 | 0.0002 | 1.78 |
| 25% | 0.002 | 0.95 | 0.001 | 2.18 |
| 50% | 0.01 | 1.11 | 0.01 | 2.32 |
| 75% | 0.03 | 1.23 | 0.02 | 2.41 |
| 95% | 0.17 | 1.50 | 0.13 | 2.90 |
| max | 0.98 | 5.15 | 0.81 | 4.45 |

* The individual arithmetic (geometric) means, CVs and GSDs were computed from the 1500 realizations for the individual.

† The population distributions ae based on the individual arithmetic(geometric) means, CVs and GSDs..

**Fig 5. Distribution of uncertainties of individual total marrow and stomach dose at the end of follow-up.**

exposures, the best fitting distributions were lognormal for more than 95% of the cohort members. For external exposures arising from EURT exposures only about 40% of the distributions could be described as lognormal and for 15% of the people scaled beta distributions provided the best fit. Except for doses arising from EURT internal exposure, none of the distributions considered appeared to be appropriate for 25% or more of the cohort members.

Examples of individuals for which the best fitting dose distribution was lognormal are shown in Fig 2B and 2D for total external dose and Fig 4C, 4D and 4H for total internal dose. The Techa internal dose for Case 4 (Fig 4G) is an example in which none of the candidate distributions considered adequately described the empirical distribution. Typically, the K-S P-values for the hypothesis of lognormality were between 0.3 and 0.9. For both marrow and stomach total doses, the individual dose realization distributions were best described as lognormal for about 70% of the cohort members. For cases in which a best-fitting distribution was chosen, the hypothesis of lognormality could rarely be rejected at the P < 0.05 level.

## Discussion

The Techa River Dosimetry System (TRDS) has been developed over several decades to provide dose estimates for people exposed from Mayak's releases of radioactive waste into the Techa River in the early 1950s and from radioactive fallout in the EURT following the 1957 explosion of a Mayak waste storage tank. Earlier versions of the TRDS provided deterministic estimates of individual annual doses arising from both internal and external exposures with little useful information on the uncertainty of these estimates. Uncertainty in dose estimates is a potential source of systematic error (bias) and increased uncertainty in the assessment of radiation risks. As described in [7, 14, 21, 23, 24], we have refined and improved the models and methods used to estimate organ doses resulting from Techa River and EURT exposures. As part of this effort, we have endeavored to characterize the uncertainty in the individual dose estimates. This was done by determining the major sources of shared and unshared uncertainties in the dose estimates, specifying distributions for these uncertainties, and using these distributions to develop multiple realizations of individual annual organ doses for 48,036 people in the combined Techa River and EURT cohorts. These distributions and the resulting 2-dimensional Monte-Carlo dosimetry system [15] have been described in this paper.

While $^{90}$Sr measurements that help reduce uncertainty in internal dose estimates are available for many cohort members with Techa River exposures, for most of the factors leading to uncertainties in individual internal and external dose estimates identified in Table 1 the nature of the uncertainty distributions was based on expert judgment. As part of our efforts to validate both the external doses and their uncertainties, we compared the individual TRDS-2016MC mean total cumulative doses to individual dose estimates based on tooth enamel Electron Paramagnetic Resonance (EPR) and Fluorescent In-Situ Hybridization (FISH) on lymphocytes [26, 27]. There was good agreement between these measurement-based dose estimates and mean TRDS16-MC estimates. The EPR results could also be used to assess the uncertainty of external radiation doses [28]. For the 220 people with EPR-based dose estimates, the 90% CI for the EPR-based estimate was compared to the 90% CI TRDS-2016MC external dose estimates.

Looking at the comparison of cumulative probability distributions (cdf) of individual total marrow and stomach doses (Fig 6A and 6B), one can see that while the range of the individual GSDs is relatively large for both marrow and stomach dose, the range is somewhat smaller for stomach than for marrow dose. The main contributor to marrow dose uncertainty, especially for cohort members with no individual $^{90}$Sr body burden measurements, is the shared uncertainty (within villages) in the intake function. On the other hand, while internal exposures

**Table 5. Distribution (%) of the best fitting cumulative distribution function for individual dose realizations.**

| Exposure | | Distribution* | | | |
|---|---|---|---|---|---|
| Type | Source | Lognormal | Scaled beta | Other | None† |
| | | Marrow | | | |
| External | Techa | 69 | 0.6 | 1.1 | 29.3 |
| | EURT | 40 | 13.2 | 2.6 | 44.2 |
| Internal | Techa | 67.1 | 0.1 | 0.3 | 32.5 |
| | EURT | 97.3 | 0 | 0.2 | 2.5 |
| Total | Techa + EURT | 72.4 | 0.1 | 1.3 | 26.2 |
| | | Stomach | | | |
| External | Techa | 67 | 0.7 | 1.2 | 31.1 |
| | EURT | 39 | 14.1 | 2.5 | 44.4 |
| Internal | Techa | 67.5 | 0.5 | 2.9 | 29.1 |
| | EURT | 96.6 | 0 | 0.1 | 3.3 |
| Total | Techa + EURT | 67.5 | 2.7 | 4.8 | 25 |

*' The candidate distributions considered were lognormal, three-parameter (scaled) beta, generalized extreme value, gamma, generalized logistic, and Weibull. The best-fitting distribution was determined using one-sample Kolmogorov-Smirnov tests and defined as the distribution with the largest P-value given that P was greater than 0.05

† The distribution was classified as none when the P-values for each of the candidate distributions was less than 0.05.

contribute to the uncertainty of the stomach dose (and other soft tissue dose) estimates (which use the same intake functions as marrow doses), these doses also depend, to a large extent, on uncertainties in the external exposures. Perhaps the marked contribution of unshared uncertainty in the behavior parameters related to time spent in various locations as a function of age and location ($T_1(\tau_i(y)), T_2(\tau_i(y))$ and $T_3(\tau_i(y))$) masks some of the shared uncertainties in the system. This could explain why the variability in the individual stomach dose GSDs are smaller than those for individual marrow doses.

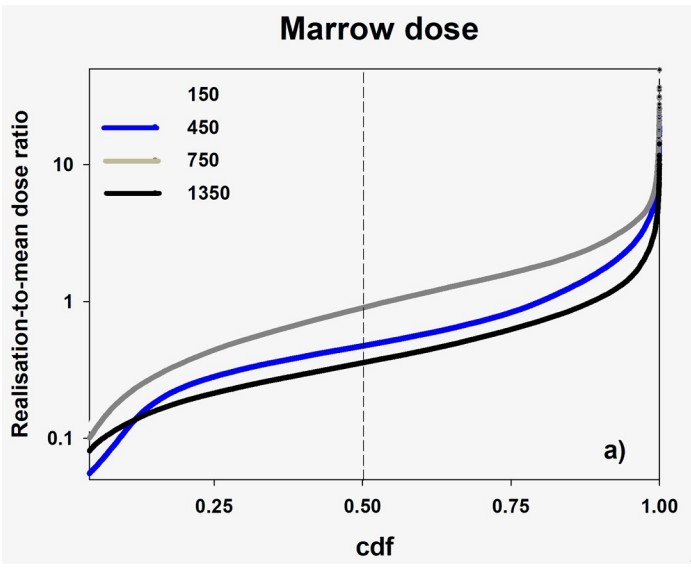
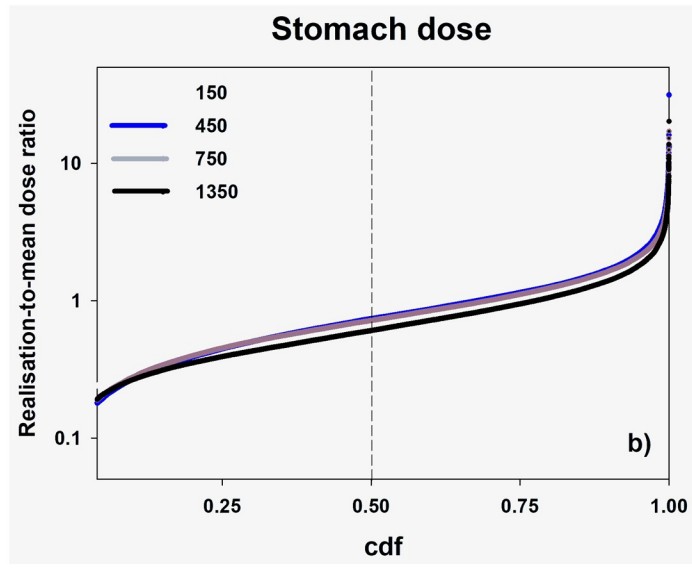

**Fig 6. Comparison of the cumulative probability distribution functions (cdf) for the ratio of the individual estimates of total marrow and stomach doses to the individual population mean doses for four dose realizations.**

It should also be noted that internal dose individualization with *IMRs* or *HSRs* is available for about 40% of the members of the combined cohort. The marrow dose uncertainties for these people are less than those for the 60% of the cohort whose marrow doses depend on the relatively large uncertainties that arise from the potential bias (shared uncertainty) in the village average *IMRs*.

Shared uncertainties result in systematic differences in the dose distributions for different dose realizations. This is illustrated in Fig 6, which presents the cumulative distribution functions of the ratio of the realization-specific total cumulative marrow (left plot) and stomach (right plot) doses to the mean organ-specific population dose for four realizations. Because of shared uncertainties within each realization, some realizations are systematically higher (150 and 450 for marrow dose in this figure) than others. As shared uncertainty increases, the between-realization spread of these cdfs will increase. The plots suggest that given the uncertainty parameters used in TRDS-2016MC, shared uncertainty is more important for marrow than for stomach doses.

As interest in assessing the effects of dose uncertainty on risk estimate uncertainty has increased, Monte-Carlo dosimetry systems have been developed for a number of studies. Table 6 summarizes the results of uncertainty estimation for some of these populations, including: people who lived near the Semipalatinsk nuclear test site (Kazakhstan) [4]; Belarusian [6] and Ukrainian [29] children, exposed to $^{131}$I as a result of the Chornobyl accident; the combined TRC/EURTC cohort; and members of the TRC exposed to $^{131}$I due to airborne releases from Mayak [30].

The first study includes 2,994 villagers under the age of 21 years old at the time of exposure (between August 1949 and September 1962) who lived downwind from the Semipalatinsk Nuclear Test Site in Kazakhstan. Individual thyroid doses from external and internal radiation sources were reconstructed from fallout deposition patterns and residential history data and dietary history derived from interviews of subjects, family members and focus groups. The fundamental mathematical dose assessment models for exposure to radioactive fallout [4] were used to simulate multiple population dose sets using a two-dimensional Monte Carlo dose estimation method accounting for shared and unshared uncertainties in dose estimation. No direct measurements of thyroid burden were available.

Similar theoretical dose reconstruction was done for the study of thyroid internal and external exposure using the limited data on atmospheric releases by PA Mayak between 1948 and 1972 considering available meteorological data for $^{131}$I transport modeling as well as the information about residence history and prolonged age-dependent dietary intakes [30]. The calculations were done for the general population (individuals of different ages) included in the Techa River cohort. The doses due to $^{131}$I were combined with the contribution of environmental contamination and dietary intakes of other radionuclides. It should be noted, the non-iodine doses to the thyroid were comparable to those for stomach described above. The mean individual iodine doses were as high as 7 Gy for residents who were born in the villages near the Mayak plant in 1948.

The second [6] and third [29] studies listed in Table 6 involve residents of the most contaminated territories of Ukraine and Belarus with radioactive fallout from Chornobyl accident, who were born before the fallout and were less than 18 years old at the time of the accident. Internal thyroid doses were reconstructed based on the direct measurements of thyroid burden in April-June 1986 and personal information on residence and diet histories. These two cohorts are of similar size and have similar dose and uncertainty sources. Individual measurements of $^{131}$I in thyroid reduce the impact of shared uncertainties. Therefore, one can expect the impact of dose uncertainties on risk analysis for these cohorts to be negligible [31–34]. In contrast, the effect of shared uncertainties is expected to be larger for the other studies listed in

**Table 6. Dose estimate uncertainties in studies with Monte-Carlo dosimetry systems.**

| Population | Target organ | Cohort size | Mean cohort dose, mGy | Mean of individual GSD over dose realizations |
|---|---|---|---|---|
| **Environmental Exposures** | | | | |
| Children residing near the Semipalatinsk nuclear test site (Kazakhstan) [4] | Thyroid | 2,376 | 280 | $\sim 3.5$* |
| Cohort of Belarusian children exposed to $^{131}$I [6] | Thyroid | 11,732 | 680 | 1.76 |
| Cohort of Ukrainian children exposed to $^{131}$I [29] | Thyroid | 13,204 | 650 | 1.6 |
| General population of the Ural region, irradiated because of the activities of the Mayak plutonium production complex | Marrow (current study) | 48,036 | 210 | 2.9 |
| | Stomach (current study) | | 40 | 2.3 |
| | Thyroid [30] | 29,735 | 200 | 2.2 |
| **Occupational Exposures** | | | | |
| US Radiologic Technologists [31] † | Colon | 110,374 | 16 | 2.1 |
| | Marrow | | 9 | 2.0 |
| Mayak Workers[5, 32] † | Lung–internal Pu | 8,340 | 187 | 2.8 |
| | Lung–external gamma | 25,940 | 418 | 1.3 |

* Estimate is an approximation based on figures in the relevant paper.

† Mean and Geometric standard deviation (GSD) estimates personal communication (Dale Preston)

Table 5. The mean thyroid doses estimated in the Urals region are smaller than those in Belarusian and Ukrainian cohorts and comparable with doses in the Kazakhstan study. Nevertheless, the thyroid dose uncertainties in [30] take an intermediate position between the uncertainties of doses estimated based on direct measurements [3, 35] and reconstructed from very limited data [4].

The relatively narrow uncertainty range of the Mayak worker external doses results because that cohort of workers all had detailed individual dosimeter readings available.

Only a few health effect studies have attempted to account for complex uncertainty in dose reconstruction by considering mixtures of shared and unshared errors [4, 5, 33, 34–37]. These studies use various approaches to calculating confidence intervals of risk factors of interest. The simplest, and least satisfactory, approach is a naïve analysis in which the risk estimates are based on analyses using either individual deterministic dose estimates (based on the "best" estimates of the various parameters in the dosimetry system) or the mean of the individual doses over dose realizations with no adjustment for dose uncertainty.

Kwon et al. [38] proposed a Bayesian model averaging (BMA) method. This method obtains a posterior probability of the slope of the dose-response, by considering all dose realization vectors produced by the 2DMC dose reconstruction system, evaluating the goodness of fit of each dose vector with the specified disease outcome, and using a Markov Chain Monte-Carlo (MCMC) method to obtain a posterior distribution for the parameter(s) of interest. Kwon and colleagues applied this method to the Semipalatinsk thyroid study [4].

Zhang et al. [2] proposed a "corrected information method" (CIM) that was inspired by the work of Stram and Kopecky [39]. In the CIM, the primary dose response analyses are carried out using the mean of the individual annual mean doses over the dose realizations and the classic (Wald) confidence bounds are adjusted using a corrected information matrix that uses the dose realizations to estimate the between realization correlations due to shared dose errors (CIM-Wald bounds). This method was used in recent analyses of plutonium and gamma effects on lung cancer risks in the Mayak Workers Cohort [5].

Zhang et al. [2] also carried out a simulation study to assess the performance of their proposed corrected information method (CIM-Wald) to adjust risk estimate confidence intervals for dose uncertainty using realistic models for the effect of doses arising from the inhalation of plutonium aerosols, lung cancer risk models in a cohort like the Mayak worker cohort. In this study, it was shown that, in the presence of shared dose uncertainty, the unadjusted Wald bounds (i.e. confidence bounds based on the assumption of asymptotic normality of the parameter estimates) had poor coverage. For the example considered in that paper, the coverage of naïve 95% Wald bounds for the dose response parameter was about 60%, while the coverage probability for the CIM-Wald adjusted confidence bounds was 94%. It should be noted, the CIM method continues to improve.

Another approach, which is often used, is likelihood-based estimation of the confidence bounds [5]. These confidence bounds improve on the classical Wald bounds by allowing for asymmetry in the distribution of the parameter estimates. However, they do not adjust for the impact of shared dose uncertainty on the confidence bounds resulting in confidence intervals that are too narrow [2, 15].

## Conclusions

Individual doses from external and internal radiation sources were reconstructed for members of the Combined Cohort of 48,036 people exposed to radiation because of Techa River contamination resulting from Mayak operations and fallout deposition in the EURT resulting from the 1957 Mayak waste storage tank explosion. The dose reconstruction made use of measured deposition patterns on the river and in the EURT, individual residential history data, individual $^{90}$Sr body-burden measurements (with propagation of intakes to unmeasured family members), data on water and foodstuff contamination, behavioral pattern surveys, and other information. The presence of $^{90}$Sr in the radionuclide composition led to markedly higher cumulative bone marrow doses than the doses to other soft tissues.

The cohort average range of mean individual doses is 0.21 Gy (min-max: 0–6.8 Gy) for marrow and 0.04 Gy (min-max: 0–1 Gy) for stomach. Cohort average dose uncertainties in terms of CV are as follows: 160% (90%CI: 89–224%) for marrow dose and 110% (89%CI: 43–150%) for stomach dose. The mean doses and the uncertainty patterns for most other soft tissues doses are similar to those for stomach dose.

The distribution of the dose realizations for individuals are highly skewed to the right and for most people the distribution of total organ doses appears to be described reasonably as lognormal, suggesting that the GM and GSD provide a useful simple summary of the uncertainty for individual cohort members. However, these summaries do not provide any information on correlation between dose realizations, which is the primary factor affecting how dose uncertainty impacts the uncertainty in risk estimates. As noted above, various methods [2, 38, 40] are being developed to use the dose realizations to adjust risk estimates for dose uncertainty. While these methods are being used, more work is needed to develop the methods and investigate their properties (bias and confidence/credibility interval coverage).

Internal exposure to $^{90}$Sr was the source of the largest proportion of cumulative dose from internal exposure for most individuals: typically, about 84% of the marrow dose and 50% of the stomach, respectively. The contribution of shared errors to the uncertainty in internal dose was reduced because many cohort members had individual measurements of the $^{90}$Sr body-burden. The autocorrelation of shared uncertainty components increases the uncertainty of cumulative doses, therefore, the greatest population-average dose uncertainty in the combined cohort is associated with the lifetime chronic internal marrow exposures due to bone-seeking long-lived $^{90}$Sr.

Analyses of solid cancer mortality in the combined cohort using the TRDS-2016MC doses are in progress. These analyses will examine how the shared dose uncertainty in the TRDS-2016MC dose estimates affects the risk estimate confidence intervals.

The choice of shared and unshared uncertainty factors and the parameters of the distributions used to characterize these factors can impact the uncertainty in the realized doses and how these uncertainties affect risk estimate uncertainty. In developing the TRDS-2016MC system, we came to realize how challenging it can be to make decisions about the uncertainty factors and how best to characterize their distributions. In this report we have focused on our final decisions. We cannot claim that the factors that we considered or that the distributions used are optimal or even that we included all important sources of uncertainty. However, we hope that they provide a reasonable assessment of the impact of dose uncertainty on risk estimate uncertainty. Now that we have some tools for and experience in computing 2DMC dose realizations and in managing the large amounts of data produced by these systems, it will be easier to investigate the impact of different choices in the design of the system.

Furthermore, we think it would be useful and feasible for researchers who have or are developing 2DMC systems to have some joint discussions in the hope of developing and publishing basic guidelines for the development of these systems.

## Supporting information

**S1 File. The unscheduled report for the project 1.1 of the U.S.–Russia Joint Coordinating Committee on Radiation Effects Research (JCCRER) "Enhancements in the Techa River Dosimetry System" (April 2020).** The report provides details on the data and models used for the stochastic approaches in TRDS-2016MC.
(PDF)

## Acknowledgments

The authors thank Daniel O. Stram and Michael A. Smith for fruitful discussions during the preparation of this paper.

## Author Contributions

**Conceptualization:** Elena A. Shishkina, Bruce A. Napier, Marina O. Degteva.

**Data curation:** Elena A. Shishkina, Dale L. Preston.

**Formal analysis:** Elena A. Shishkina, Dale L. Preston.

**Funding acquisition:** Bruce A. Napier.

**Methodology:** Bruce A. Napier, Marina O. Degteva.

**Software:** Bruce A. Napier, Dale L. Preston.

**Supervision:** Bruce A. Napier, Marina O. Degteva.

**Visualization:** Dale L. Preston.

**Writing – original draft:** Elena A. Shishkina, Bruce A. Napier, Marina O. Degteva.

**Writing – review & editing:** Bruce A. Napier, Dale L. Preston.

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
