## [Decision Letter · Decision Letter 0]

12 Apr 2023

PONE-D-23-01650Dose uncertainties for radiation epidemiology: research experience in the Urals regioPLOS ONE

Dear Dr. Napier,

Thank you for submitting your manuscript to PLOS ONE. After careful consideration, we feel that it has merit but does not fully meet PLOS ONE’s publication criteria as it currently stands. Therefore, we invite you to submit a revised version of the manuscript that addresses the points raised during the review process. I have finally received two expert reviews and both are positive. Congratulations! We are almost there. Both reviews have made a handful of very small edits that should be addressed. Please take care of these asap and we will move the ms to the next step.  Please submit your revised manuscript by May 27 2023 11:59PM. If you will need more time than this to complete your revisions, please reply to this message or contact the journal office at plosone@plos.org. Please include the following items when submitting your revised manuscript:A rebuttal letter that responds to each point raised by the academic editor and reviewer(s). You should upload this letter as a separate file labeled 'Response to Reviewers'.A marked-up copy of your manuscript that highlights changes made to the original version. You should upload this as a separate file labeled 'Revised Manuscript with Track Changes'.An unmarked version of your revised paper without tracked changes. You should upload this as a separate file labeled 'Manuscript'.If applicable, we recommend that you deposit your laboratory protocols in protocols.io to enhance the reproducibility of your results. Protocols.io assigns your protocol its own identifier (DOI) so that it can be cited independently in the future. For instructions see: https://journals.plos.org/plosone/s/submission-guidelines#loc-laboratory-protocols. Additionally, PLOS ONE offers an option for publishing peer-reviewed Lab Protocol articles, which describe protocols hosted on protocols.io. Read more information on sharing protocols at https://plos.org/protocols?utm_medium=editorial-email&utm_source=authorletters&utm_campaign=protocols.

We look forward to receiving your revised manuscript.

Kind regards,

Tim A. Mousseau

Academic Editor

PLOS ONE

Journal Requirements:

"EAS: Federal Medical-Biological Agency of Russia Contract N◦ 27.501.19.2 in the framework of Russian Federal Targeted Program “Provision of nuclear and radiation safety for the period 2016-2020 and for the period up to 2030”. http://government.ru/en/department/497/

BAN: PNNL Contract DE-AC05-76RL01830, (US Department of Energy), Project  JCCRER DOSE RECONSTRUCTION FOR THE URALS,  Budget and Reporting Number HS0240030, https://www.energy.gov/ehss/international-health-studies-and-activities

DLP: University of Southern California Prime Award #   DE-HS0000091  (US Department of Energy), Project                                          Epidemiological and Biostatistical Assistance for Project 2.2 Mayak Worker Cancer Mortality and for Project 1.2 Techa River Cohort Cancer mortality and Incidence,  USC Subaward  122032572 – Mod 3, https://keck.usc.edu/

MOD: Federal Medical-Biological Agency of Russia Contract N◦ 27.501.19.2 in the framework of Russian Federal Targeted Program “Provision of nuclear and radiation safety for the period 2016-2020 and for the period up to 2030”. http://government.ru/en/department/497/ (deceased)"

Reviewer's Responses to Questions

**Comments to the Author**

1. Is the manuscript technically sound, and do the data support the conclusions?

Reviewer #1: Yes

Reviewer #2: Yes

2. Has the statistical analysis been performed appropriately and rigorously? 

Reviewer #1: Yes

Reviewer #2: Yes

3. Have the authors made all data underlying the findings in their manuscript fully available?

Reviewer #1: Yes

Reviewer #2: Yes

4. Is the manuscript presented in an intelligible fashion and written in standard English?

Reviewer #1: Yes

Reviewer #2: Yes

5. Review Comments to the Author

Reviewer #1: This paper considers dose uncertainties in the TRDS-2016MC. Individual doses from external and internal radiation sources were reconstructed for 48,036 people based on environmental contamination patterns, residential histories, individual 90Sr body-burden measurements and dietary intakes. The paper is very well and clearly written and will present substantial interest to PLOS One audience. I have comments and suggestions (please see attached file) that, I believe, will help to improve the manuscript.

Reviewer #2: This is an impressive piece of work. I have only suggestions for minor revisions, which are included in the accompanying annotated manuscript. I look forward to epidemiological analyses that incorporate the dosimetry uncertainties described in the manuscript.

6. PLOS authors have the option to publish the peer review history of their article (what does this mean?). If published, this will include your full peer review and any attached files.

Reviewer #1: No

Reviewer #2: No

---

## [Author Response · Author response to Decision Letter 0]

14 Jun 2023

Both reviewers provided their comments on pdf copies of the submitted file. In what follows we refer to the page numbers lines numbers provided by the manuscript management software and the page numbers in the manuscript file that we provided. 

Reviewer 1

We appreciate the reviewer’s careful reading of the manuscript and the helpful comments. As indicated below, we have made the suggested corrections and, as necessary, revised the text to address the reviewer’s concerns. We appreciate the suggested references and have cited them in the revised text. 

Page 3

Lines 70- 71 : the text was modified to read “underestimation of the risk estimate uncertainty”

Page 4

FPSGV-HU9YN-9EH6Q-SE7LX-47HXCPage 5

Line 103: “Techa River Dosimetry System” is deleted and replaced by TRDS-2016

Line 106 : In general, there are currently no guidelines for setting the number of realizations in a MC dosimetry system. However, in this case the number was chosen in part because of the number of parameters in the dose uncertainty model and the requirement of at least one realization per parameter for the Latin hypercube sampling algorithm used . 

Line 110: changed as suggested to TRC and EURT

Pages 6-7

Lines 123-124 Omitted the phrase for 23 organs … as 67 years as suggested.

Lines 144 -149 Omit bullets also drop “while” at beginning of line 149

Line 151 Corrected text to I_y^( _ ^90 Sr ) as in Eqn 1

Line 159 The reference to ONIS was dropped since this level of detail is of no detail almost all readers. 

Pages 9 10

Lines 201-207 & 225-226 replaced -with “is” as requested

Line 226 The text was corrected to read “is a conversion factor from” …

Page 11

Line 236 Equation 5 was corrected as suggested

Line 239 The text was modified to read “this parameter has the same value as for the Techa River”

Line 240-241 “the same as for the Techa River “is deleted”

Line 242 We have added a short description of how the individual and population statistics were computed where the population and individual statistics are first discussed (line 309) and in the notes to Tables 2 and 3.

Line 250 “..for both internal and external exposure” was added as suggested

Page 12

Lines 258 and 260 reworded to be more direct along the lines suggested 

Line 272 (Table 1) .The table has been modified.

Page 17

Line 294 Changed to acronyms, as suggested.

Lines 309 & 311 We have added information on the meaning of population and individual statistics here and in the notes to Tables 2 and 3..

Line 314 We have now given the median with two significant digits.

Page 18

line 327 We modified the text to indicate that the short-lived radionuclides include all radionuclides other than 137Cs. 

Page 22

Lines 414 & 416 Replaced 25th percentiles with medians (18 and 174) 

Page 28

Line 546+(Table 4) We have revised the table title and added notes explaining the difference between population and individual statistics. 

Page 33-34. 

Line 677+ and 699 (Table 6) References and medians updated as suggested. 

Reviewer 2

We are grateful for the positive comments, the suggested corrections and references. We have revised the text to address all of the issues raised.

Page 2.

Line 24. Changed to “Russian Southern Urals”

Page 4. 

Line 79 Changed to “… populations of the Russian Southern Urals …”

Line 83 And 85. The suggested references were added.

Line 84 added (sometimes called the “Kyshtym Accident”) 

(which is somewhat ironic since the accident occurred at Mayak and Kyshtym was not really in the contaminated area)

Page 5. 

Line 117. Changed to “bone marrow”

Page 6 

Lines 135-136 modified to Techa River internal exposures

Page 9. 

Line 191 modified to Techa River External exposures

Page 13. 

Line 284 changed to active bone marrow

Page 14. 

Line 288 Changed to Techa River Dosimetry System (TRDS)

Page 25 

Line 452 AM changed to active bone marrow

Page 34 

Line 695 change to non-iodine

Page 35 

Line 752 changed to … contamination resulting from Mayak operations and fallout deposition in the EURT resulting from the 1957 Mayak waste storage tank explosion

Page 38 

Lines 829 and 832 Suggested references were added

---

## [Editor Report · Decision Letter 1]

28 Jun 2023

Dose estimates and their uncertainties for use in epidemiological studies of radiation-exposed populations in the Russian Southern Urals

PONE-D-23-01650R1

Dear Bruce,

Sorry for the short delay. I am just back from a month's field work in the polygon and Chernobyl.... 

We’re pleased to inform you that your manuscript has been judged scientifically suitable for publication and will be formally accepted for publication once it meets all outstanding technical requirements.

This is an important piece of work and I congratulate you on carrying it through. Personally, being geographically oriented, I would have included a simple map showing the study region in relation to some of the other key sites and cities in the region. But perhaps this is just me! 

Kind regards,

Tim A. Mousseau

Academic Editor

PLOS ONE

Professor of Biological Sciences, University of South Carolina

---

## [Editor Report · Acceptance letter]

31 Jul 2023

PONE-D-23-01650R1 

Dose estimates and their uncertainties for use in epidemiological studies of radiation-exposed populations in the Russian Southern Urals. 

Dear Dr. Napier:

I'm pleased to inform you that your manuscript has been deemed suitable for publication in PLOS ONE. Congratulations! Your manuscript is now with our production department. 

Kind regards, 

on behalf of

Dr. Tim A. Mousseau 

Academic Editor

PLOS ONE